# Early tissue damage and microstructural reorganization predict disease severity in experimental epilepsy

**Philipp Janz[1,2†], Niels Schwaderlapp[3†], Katharina Heining[2,4†], Ute Häussler[1,5], Jan G Korvink[6], Dominik von Elverfeldt[3], Jürgen Hennig[3,5], Ulrich Egert[4,5,7], Pierre LeVan[3,5‡], Carola A Haas[1,5,7*‡]**

[1]Experimental Epilepsy Research, Department of Neurosurgery, Medical Center – University of Freiburg, Faculty of Medicine, University of Freiburg, Freiburg, Germany; [2]Faculty of Biology, University of Freiburg, Freiburg, Germany; [3]Medical Physics, Department of Radiology, Medical Center – University of Freiburg, Faculty of Medicine, University of Freiburg, Freiburg, Germany; [4]Laboratory for Biomicrotechnology, Department of Microsystems Engineering, University of Freiburg, Freiburg, Germany; [5]BrainLinks-BrainTools Cluster of Excellence, University of Freiburg, Freiburg, Germany; [6]Institute of Microstructure Technology, Karlsruhe Institute of Technology, Karlsruhe, Germany; [7]Bernstein Center Freiburg, University of Freiburg, Freiburg, Germany

**\*For correspondence:** carola. haas@uniklinik-freiburg.de

[†]These authors also contributed equally to this work
[‡]These authors also contributed equally to this work

**Competing interests:** The authors declare that no competing interests exist.

**Abstract** Mesial temporal lobe epilepsy (mTLE) is the most common focal epilepsy in adults and is often refractory to medication. So far, resection of the epileptogenic focus represents the only curative therapy. It is unknown whether pathological processes preceding epilepsy onset are indicators of later disease severity. Using longitudinal multi-modal MRI, we monitored hippocampal injury and tissue reorganization during epileptogenesis in a mouse mTLE model. The prognostic value of MRI biomarkers was assessed by retrospective correlations with pathological hallmarks Here, we show for the first time that the extent of early hippocampal neurodegeneration and progressive microstructural changes in the dentate gyrus translate to the severity of hippocampal sclerosis and seizure burden in chronic epilepsy. Moreover, we demonstrate that structural MRI biomarkers reflect the extent of sclerosis in human hippocampi. Our findings may allow an early prognosis of disease severity in mTLE before its first clinical manifestations, thus expanding the therapeutic window.

## Introduction

Epilepsy is a devastating disease, with a prevalence of about 1%, which makes it one of the most common neurological disorders. Of particular clinical significance is mesial temporal lobe epilepsy (mTLE), in which seizures arise from mesial temporal limbic structures, as it is the most frequent form of epilepsy in adults (40%) and particularly resistant to pharmacological treatment (*Engel, 2001*). Hippocampal sclerosis (HS), characterized by the loss of CA1 and CA3 pyramidal cells, gliosis and associated with granule cell dispersion (GCD), represents the most common pathological hallmark of intractable mTLE (*Thom, 2014*; *Walker, 2015*). HS can emerge secondary to an initial precipitating insult, e.g. status epilepticus (SE), head trauma, febrile seizures or limbic encephalitis (*Engel, 2001*). To date, resection of the lesioned, epileptogenic tissue is the only therapeutic intervention. However, not all patients remain seizure-free (*Ryvlin and Kahane, 2005*). In this context, non-invasive

**eLife digest** Roughly one percent of people in the world suffer from epilepsy, a disorder in which individuals experience seizures due to abnormal electrical activity in the brain. Seizures can vary from brief episodes of amnesia or déjà-vu to convulsions and loss of consciousness. In adults, the most common form of epilepsy is known as temporal lobe epilepsy. As the name suggests, this type of epilepsy originates in a region of the brain called the temporal lobe, usually within a structure called the hippocampus.

Many patients who develop temporal lobe epilepsy will have experienced a head injury or infection earlier in life that damaged their hippocampus. However, damage to the hippocampus does not always lead to epilepsy. Moreover, many years may pass between the damage and the onset of regular seizures. While some patients find that anti-epileptic drugs can control their seizures, others experience no benefit. For these patients, the only effective treatment is to remove the damaged brain tissue.

At present, there is no way of knowing which patients with a damaged hippocampus will go on to develop temporal lobe epilepsy. To identify the deciding factors, Janz, Schwaderlapp, Heining et al. treated mice with a toxin that can damage the hippocampus. After roughly two weeks most of the mice were experiencing regular seizures. Imaging the animals' brains during this two week period revealed that mice whose hippocampi showed more severe cell death shortly after exposure to the toxin surprisingly developed a milder form of epilepsy. The same was also true for animals whose hippocampi showed signs of being extensively reorganized. Further experiments show that samples of hippocampal tissue from the brains of human patients with temporal lobe epilepsy also showed these same cellular features.

The next step is to test whether these changes can be used to predict which patients with hippocampal damage will develop epilepsy later in life. Identifying at-risk individuals would allow them to be treated earlier and hopefully prevent them from developing epilepsy in the first place.

MRI techniques have become an important method to identify sclerotic tissue as a biomarker in both human patients (*Gomes and Shinnar, 2011*; *Urbach et al., 2014*) and rodent TLE models (*Nehlig, 2011*; *Shultz et al., 2014*; *Sierra et al., 2015a*). Up to now, this has been, however, restricted to the chronic stage of epilepsy when seizures have already emerged.

A promising notion is to start medical treatment during early epileptogenesis before the first seizure arises. This stands in contrast to current anti-epileptic treatments that target clinical symptoms, namely the seizures themselves, but do not cure the underlying disease. However, progress in this direction is impeded by the impossibility to identify patients before they develop epilepsy and experience their first seizure. In mTLE patients, the period between the initial epileptogenic insult and chronic epilepsy spans many years (*Lukasiuk and Becker, 2014*), severely hindering the search for and validation of predictive biomarkers in human studies. Conversely, in animal models, epileptogenesis can be induced within weeks to months (*Lévesque and Avoli, 2013*), providing an excellent opportunity to identify putative anatomical and physiological biomarkers of epileptogenesis.

Considering that multiple cellular and molecular processes accompany the progression of HS, it may be possible to image early pathological changes during epileptogenesis that may prospect the later state. So far, a few MRI biomarkers have been identified that indicate epilepsy onset or seizure susceptibility secondary to pilocarpine-induced SE (*Roch et al., 2002*; *Choy et al., 2010*), traumatic brain injury (*Kharatishvili et al., 2007*; *Immonen et al., 2013*) or prolonged febrile seizures (*Choy et al., 2014*). However, since distinct unilateral HS is lacking in these animal models, prognostic MRI biomarkers for the most common etiology of pharmacoresistant epilepsy have not been identified yet.

To address this issue, we applied longitudinal multi-modal MRI in the intrahippocampal kainate mouse model that replicates the major features of human mTLE (i.e. the development of HS and of spontaneous recurrent seizures during epileptogenesis; *Bouilleret et al., 1999*; *Riban et al. (2002)*; *Heinrich et al., 2011*). We monitored distinct features of the pathogenesis focusing on the hippocampus (*Figure 1—figure supplement 2*), which is mainly affected in pharmacoresistant mTLE

(*Malmgren and Thom, 2012*; *Cendes et al., 2014*; *Walker, 2015*). We performed a retrospective correlation of MRI measures with structural changes identified by immunohistology, and determined the prognostic value of imaging biomarkers with respect to the severity of HS. Moreover, we applied multivariate data analysis to evaluate and compare the performance of individual biomarkers to predict also the seizure severity. Our study demonstrates for the first time that the extent of epileptogenesis-associated tissue alterations in the hippocampus directly mirrors the ensuing severity of intractable mTLE.

## Results

### Inter-individual variability of histological changes associated with mTLE

Little is known about the variability of histopathological changes across kainate-injected mice; yet this information is critical to relate early imaging biomarkers to HS severity in mTLE. Therefore, we quantified the degree of histopathological features associated with HS (i.e. cell death-associated microgliosis and GCD) in all injected mice individually (*Figure 1*).

The extent of GCD (*Figure 1A–C*) and cell death-associated microgliosis (*Figure 1D,E*) varied substantially across kainate-injected mice ($n_{KA}$ = 8), but was positively correlated with each other (*Figure 1F*; $r^2$ = 0.69, p<0.01). In addition, intrahippocampal EEG recordings validated the presence of epileptiform activity in all kainate-injected mice (*Figure 1G*). However, one kainate-injected mouse (NP10) had no paroxysmal discharges, even though it exhibited single epileptic spikes, and therefore it was considered as non-epileptic. All epileptic mice showed robust GCD indicated by an increase in the total GCL volume compared to non-epileptic mice (*Figure 1H*; *non-epi.*: 415.8 ± 9.6 * $10^6$ μm$^3$; *epi.*: 1379.0 ± 164.5 * $10^8$ μm$^3$, p<0.01) and a high spatial extent of cell death-associated microgliosis (*Figure 1I*; *non-epi.*: 563.8 ± 493.7 μm; *epi.*: 6965.0 ± 811.8 μm, p<0.001).

To monitor the temporal development of epileptic activity another set of mice was implanted with electrodes directly after kainate injection and recorded at different time points during 2 (n = 3) or 3 weeks (n = 5) of disease progression (*Figure 1—figure supplement 1*). Consistent with previous studies in the same animal model (*Riban et al., 2002*; *Arabadzisz et al., 2005*; *Heinrich et al., 2011*), we observed prolonged seizures during SE, followed by a period exhibiting mainly isolated epileptic spikes and low-amplitude bursts. The first high-amplitude recurrent paroxysmal episodes were recorded consistently after approximately two weeks (*Figure 1—figure supplement 1H*; number of epileptic discharges, *1d*: 0.014 ± 0.004; *0.5w*: 0.002 ± 0.002; *1w*: 0.036 ± 0.019; *2w*: 0.240 ± 0.065, p<0.01 vs 1w; *3w*: 0.274 ± 0.070, p<0.01 vs 1w). In only two out of eight mice regular paroxysmal episodes were already evident at one week after SE. In conclusion, this shows that although the speed of disease progression varies among mice, the chronic epileptic stage is reached within the second week after SE in this animal model.

### Early neuronal cell death determines disease severity in chronic epilepsy

Considering the inter-individual histopathological differences observed in kainate-injected mice during the chronic stage of mTLE, we investigated the extent to which high-resolution $T_2$-weighted imaging reflects SE-induced hippocampal injury and whether observed early tissue damage and anatomical changes during epileptogenesis might predict the subsequent HS severity in chronically epileptic mice (*Figure 2*).

$T_2$-weighted imaging revealed a biphasic rise in hippocampal $T_2$ intensity, characterized by a transient increase at day 1, followed by a drop to baseline at day 4 and a subsequent increase at 8 days after SE (*Figure 2A,B*). The early increase of $T_2$ intensity was observed in the dentate gyrus (DG) (*Figure 2I*; *1d:* 10.6%) and the CA1 region (*Figure 2G*; *1d:* 26.9%; both p<0.001), whereas in the late phase it was restricted to the DG (*8d*: 7.8%; *31d*: 7.9%, both p<0.05). Accordingly, in chronically epileptic mice both regions showed distinct histopathological changes. Extensive death of CA1 pyramidal cells was associated with dense microglial scarring (*Figure 2F,F1*). Moreover, hilar interneurons (*Figure 2F,F3*) were also lost. Dentate granule cells, however, remained intact but dispersed broadly (*Figure 2F2*). Importantly, 1 day following SE, $T_2$ intensity in the CA1 region significantly correlated with cell death-associated microgliosis (*Figure 2H*; $r^2$ = 0.88, p<0.001). Similarly, in the DG both early (at 1 day) and subsequent $T_2$ hyperintensities (at 8–31 days) were related to the later

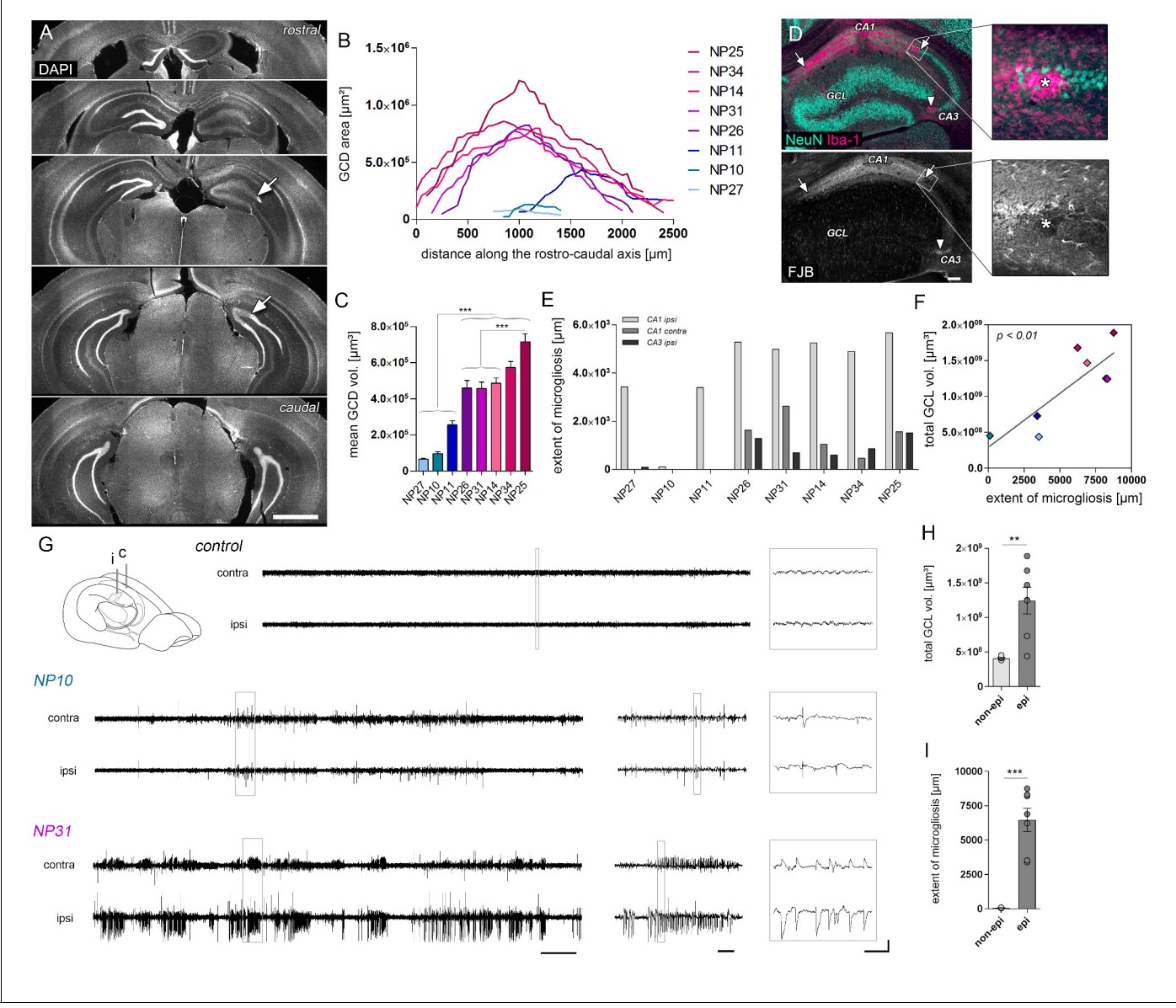

**Figure 1.** The severity of histological changes associated with HS varies in chronically epileptic mice. (A) Representative DAPI-stained sections at different levels along the rostro-caudal hippocampal axis showing the extent of GCD (arrow indicates the transition to non-dispersed regions). (B) Corresponding quantitative analysis of the GCD area along the rostro-caudal axis, and (C) the calculated mean GCD volume from all analyzed sections tested for individual kainate-injected mice (one-way ANOVA, Bonferroni's post-test; ***p<0.001; n = 8). (D) Representative photomicrographs of NeuN (turquoise; neurons) and Iba-1 (magenta; microglia) double immunostaining (upper panel) and Fluoro-Jade B (FJB) staining in consecutive sections. Clusters of amoeboid microglia are tightly associated with FJB-stained dying neurons (arrows and asterisks). (E) Quantitative analysis of cell death-associated microglial scarring in different regions (CA1 ipsi and contra; CA3 ipsi). (F) Regression analysis for the degree of GCD and the extent of microgliosis (summed for all regions) in kainate-injected mice (n = 8; Pearson's correlation). Kainate-injected mice (NP10, NP11, NP14, NP26, NP27, NP31, NP34) are color-coded. Scale bars in A, 1 mm; in D, 200 µm. (G) Schematic of the mouse brain adapted from *Witter and Amaral, 2004*. Representative EEG traces of non-epileptic mice (controls or mice displaying only single epileptic spikes, NP10) and one example of an epileptic mouse displaying both epileptic spikes and paroxysmal discharges (NP31). Horizontal scale bars (left) 50 s, (middle) 5 s, (right) 0.5 s; vertical scale bar 2 mV. (H–I) Quantitative analysis of the total GCL volume (summed for all analyzed sections) and extent of microgliosis for epileptic (dark grey) and non-epileptic mice (light grey), respectively. Student's t-test; **p<0.01, ***p<0.001; $n_{non-epi}$ = 6, $n_{epi}$ = 7. All values are presented as the mean ± SEM.

The following figure supplements are available for figure 1:

**Figure supplement 1.** Longitudinal development of epileptiform activity after kainate injection.

*Figure 1 continued on next page*

*Figure 1 continued*

**Figure supplement 2.** Schematic of the experimental design.

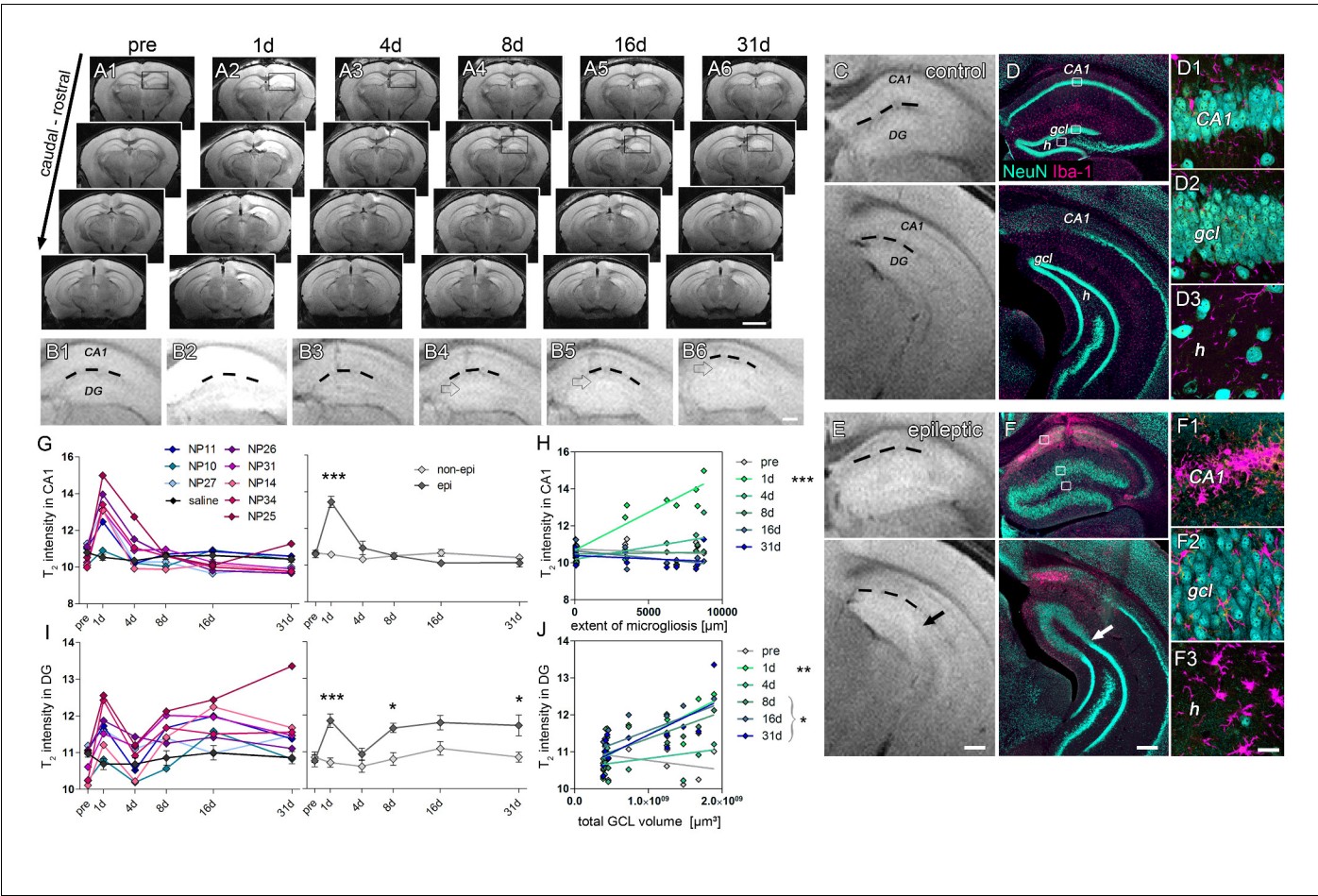

**Figure 2.** Initial hippocampal damage detected by $T_2$ imaging predicts the degree of HS. (**A1-6**) Overview of whole-brain $T_2$ intensity maps along the rostro-caudal axis before (pre) and at distinct time-points following kainate-induced SE (1d, 4d, 8d, 16d and 31d). (**B1-6**) Enlarged view of the ipsilateral hippocampus. Dashed lines denote the hippocampal fissure. Open arrows mark the GCD. (**C–D, E–F**) Direct comparison between $T_2$ images and NeuN (turquoise) and Iba-1 (magenta) double-stained sections. Upper and lower panels show the septal and temporal regions of the hippocampus, respectively. Arrows indicate the transition from the dispersed to the non-dispersed GCL. (**D1-3, F1-3**) High-magnification confocal images corresponding to photomicrographs in D and F display the loss of neurons and accompanied microglial scarring in the CA1 region and the hilus. Principal neurons in the GCL remain intact but are highly dispersed. H, hilus; gcl, granule cell layer. Scale bars: A, 2 mm; B,E,F, 200 μm; F3, 20 μm. (**G, I**) Quantitative analysis of $T_2$ changes in CA1 and the DG during epileptogenesis plotted for individual animals (left panel: controls, black, n = 5; kainate-injected mice, color-coded), and (right panel) statistically tested for the epileptic (dark grey) and non-epileptic group (light grey). Source data is provided in '***Figure 2—source data 1***'. Two-way ANOVA; Bonferroni's post-test; *p<0.05, ***p<0.001, $n_{non-epi}$ = 6, $n_{epi}$ = 7. Values are presented as the mean ± SEM. (**H, J**) Corresponding linear regression analysis of $T_2$ intensity at distinct time points during epileptogenesis (color-coded) with region-specific histopathological changes in epileptic and non-epileptic mice (n = 13; Pearson's correlation; corrected for multiple testing; *p<0.05, **p<0.01, ***p<0.001).

The following source data and figure supplement are available for figure 2:

**Source data 1.** Summary of $T_2$ metrics.

**Figure supplement 1.** $T_2$ changes in the piriform cortex and the amygdala during epileptogenesis.

degree of GCD (*Figure 2J*). No significant change of $T_2$ intensity was found in the piriform cortex or the amygdala (*Figure 2—figure supplement 1*).

The dynamics of neurodegeneration during epileptogenesis were further characterized by [1]H-MR spectroscopy (*Figure 3*). N-acetyl aspartate (NAA), serving as a surrogate marker for neurons, and the neurotransmitters glutamate and GABA were measured to differentiate between excitatory principal neurons and inhibitory interneurons, respectively. As early as 1 day following SE, the concentrations of NAA (−45.3%), glutamate (−35.1%) and GABA (−39.4%; all p<0.001) decreased substantially (*Figure 3C,E,G*) which was correlated with the later extent of cell death-associated microgliosis (*Figure 3D,F,H*; *for NAA:* $r^2$ = 0.89; *for glutamate:* $r^2$ = 0.93; *for GABA:* $r^2$ = 0.77; all p<0.001). In contrast to glutamate, GABA concentrations remained correlated also at day 4 ($r^2$ = 0.74; p<0.01), but this disappeared from 8 days onward, which was accompanied by a gradual increase of GABA above control values after 31 days. Conversely, NAA and glutamate concentrations remained low during subsequent epileptogenesis correlating with microgliosis. We also found an early upregulation of lactate and a delayed rise of myoinositol (*Figure 3I,K*), serving as markers for micro- and astroglial activation, respectively (*Nehlig, 2011*). However, changes in both metabolites showed weak correlations with chronic histopathological changes (*Figure 3J,L*).

These results show that the magnitude of initial neurodegeneration following SE, as quantified by MR biomarkers, predicts the severity of HS in the chronic stage of epilepsy.

## Dynamics of microstructural alterations in the dentate gyrus during epileptogenesis

SE-induced neurodegeneration is known to trigger various cellular responses, including inflammatory processes, synaptic plasticity and migration of surviving neurons and glial cells. To probe our hypothesis that these responses might entail microstructural alterations detectable by MRI, specifically diffusion-weighted imaging (DWI), we first identified the morphological changes of granule cells and radial glia cells, two major cell populations which survive in the sclerotic hippocampus (*Figure 4*). We used transgenic Thy1-eGFP mice, in which a subset of dentate granule cells is intrinsically labeled by eGFP, to investigate their dendritic and axonal morphology as well as the progression of GCD. Accordingly, quantitative analysis revealed a continuous increase of the GCL width at day 7, reaching significance at 14 and 21 days after SE when compared to controls (*Figure 4A,B*; *control*: 67.8 ± 2.5 μm; *14d:* 132.6 ± 13.8 μm; *21d:* 168.6 ± 9.9 μm, both p<0.001). This increase was accompanied by a strong disorganization of their distal dendrites in the molecular layer, whereas proximal dendritic segments residing within the GCL became longer and appeared more organized, possibly due to traction forces mediated by GCD. Furthermore, proximal dendrites significantly thickened as early as day 7 (*control*: 1.54 ± 0.04 μm, *7d:* 1.86 ± 0.12 μm, p<0.05), while proximal axons increased in volume only during later stages of epileptogenesis at 14 and 21 days after SE (*Figure 4C,D*; *control*: 0.70 ± 0.02, *14d:* 1.15 ± 0.08 μm; *21d:* 1.63 ± 0.09 μm, both p<0.001).

Next, we determined the time course of axonal rearrangement within the GCL, by analyzing the optical density of ZnT3-labeled mossy fibers. A few ZnT-3-labeled profiles were already evident at days 4 and 7 in the subgranular region, but a significant increase of ZnT-3 was only detected at 14 and 21 days after SE (*Figure 4E,F*; *control:* 4.02 ± 0.50, *14d:* 11.64 ± 2.39, p>0.05; *21d:* 14.57 ± 2.11, p<0.001). Although sprouting of mossy fiber collaterals is impressive, it is important to note that these axons are not uniformly aligned, limiting their detectability by DWI.

In contrast, processes of radial glia cells are known to radially transverse the GCL in a highly organized fashion (*Heinrich et al., 2006*). Accordingly, we reconstructed GFAP-labeled radial glia processes in three-dimensional space to perform quantitative morphometry. Interestingly, the mean volume of radial glia processes significantly increased as early as 4 days after SE (*Figure 4G,H,I*; *control:* 6.47 ± 1.15 μm³, *4d:* 19.09 ± 0.55 μm³, p<0.001). Compared to controls this increase was further augmented at later time points during epileptogenesis (*7d:* 20.59 ± 1.37; *14d:* 29.26 ± 4.03; *21d:* 28.75 ± 2.59, all p<0.001). Moreover, the optical density of GFAP was correlated with the size of the GCL in chronically epileptic mice (*Figure 4M*; $r^2$ = 0.89, p<0.0001), showing that radial gliosis is tightly associated with GCD.

Taken together, our results demonstrate that during epileptogenesis distinct microstructural alterations emerge at different time points within the DG: Radial gliosis starts early after SE, whereas neuronal hypertrophy, dendritic displacement and synaptogenesis accompany GCD at later stages.

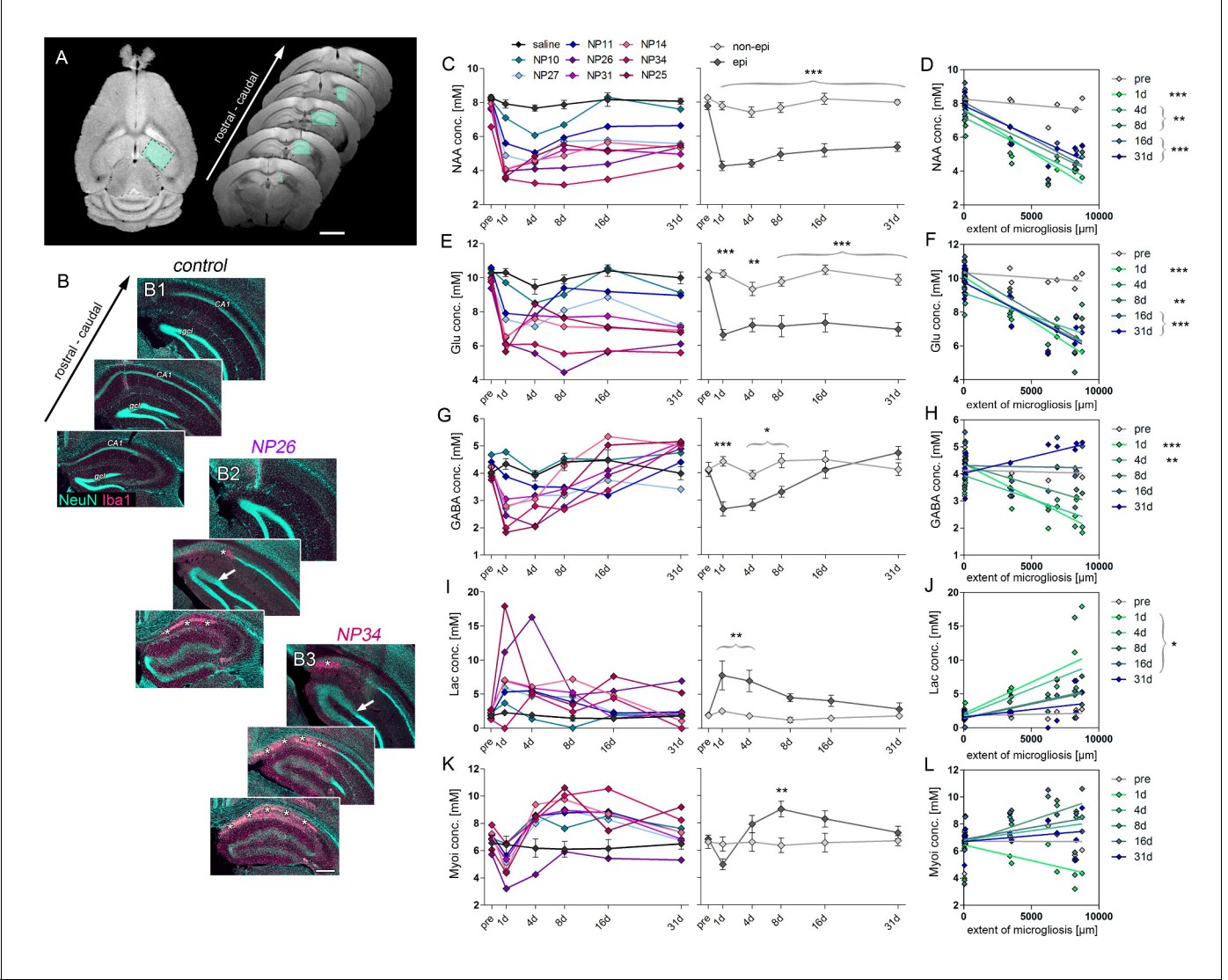

**Figure 3.** Early decline of glutamate and GABA predict HS. (A) Representative horizontal and coronal T$_2$ images illustrating the region-of-interest in the ipsilateral hippocampus (turquoise boxes) for $^1$H-MR-spectroscopy. N-acetyl aspartate (NAA) served as a marker for neurons. Glutamate (Glu) and gamma-aminobutyric acid (GABA) allowed to estimate the loss of excitatory and inhibitory neurons, respectively. Lactate (Lac) and myoinositol (Myoi) were used as surrogate markers for microglial and astroglial activation. (B) Representative photomicrographs of NeuN (turquoise, neurons) and Iba-1 (magenta, microglia) double-stained sections from one saline- (control) and two kainate-injected mice exhibiting different degrees of hippocampal sclerosis (NP26, moderate; NP34, strong) for qualitative comparison with the degree of metabolic alterations. Arrow, borders of the GCD; Asterisks, cell loss and microglial scarring in CA1. Scale bar in A, 2 mm; B, 200 μm. (C, E, G, I, K) Quantitative analysis of NAA, Glu, GABA, Lac and Myoi concentrations plotted for individual mice (left panel; controls, black, n = 5; kainate-injected animals color-coded) and groups (right panel; epileptic, dark grey; non-epileptic, light grey) during epileptogenesis. Source data is provided in '*Figure 3—source data 1*'. Two-way ANOVA; Bonferroni's post-test; *p<0.05, **p<0.01, ***p<0.001; n$_{non-epi}$ = 6, n$_{epi}$ = 7. Values are presented as the mean ± SEM. (D, F, H, J, L) Corresponding linear regression analysis of metabolite concentrations at distinct time-points during epileptogenesis (color-coded) with the extent of cell death-associated microgliosis in epileptic and non-epileptic mice (n = 13; Pearson's correlation, corrected for multiple comparison; *p<0.05, **p<0.01, ***p<0.001).

The following source data is available for figure 3:

**Source data 1.** Summary of $^1$H-MR metrics.

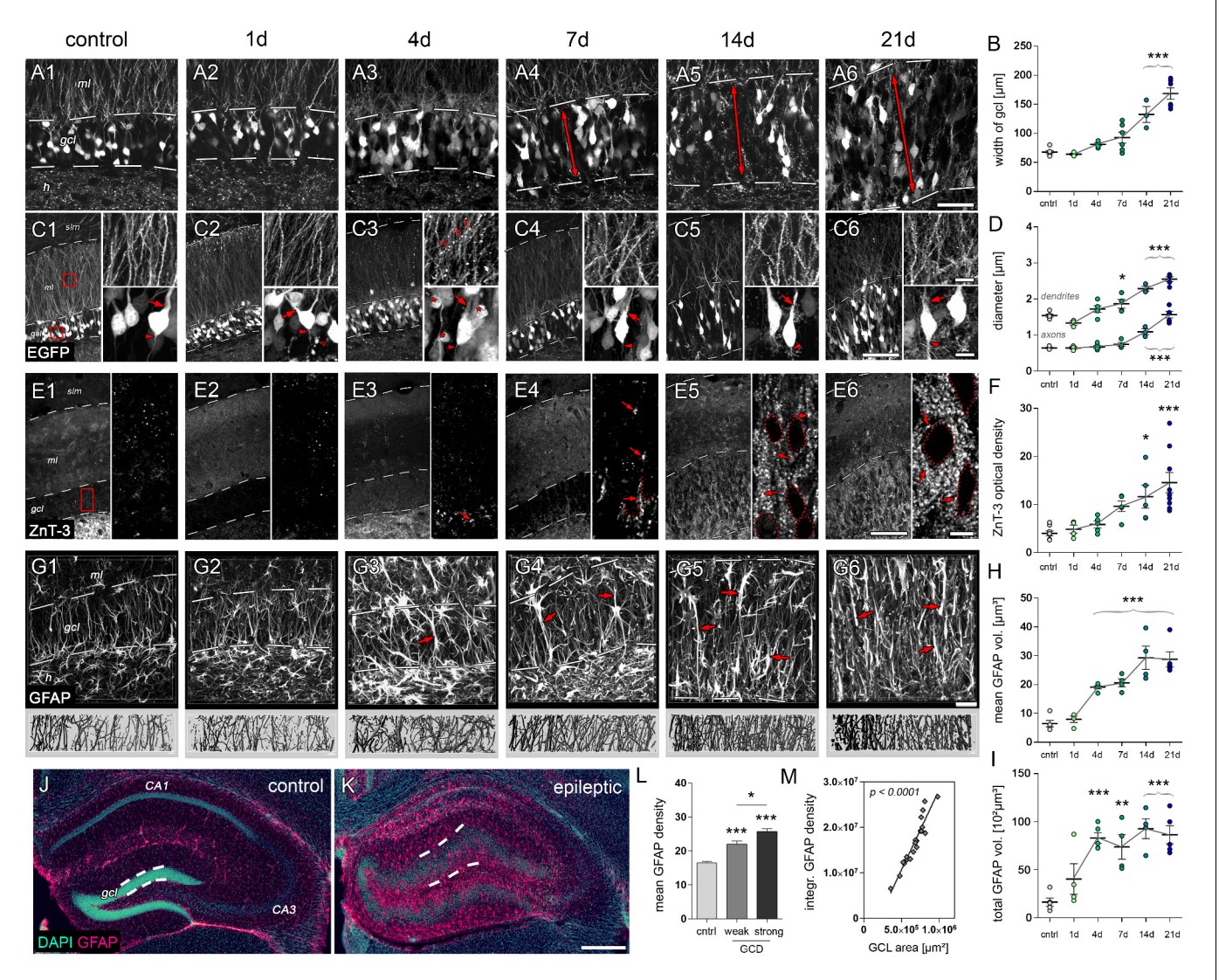

**Figure 4.** Microstructural alterations in the DG during epileptogenesis. (**A1-6** and **C1-6**) Representative confocal images show changes in the cytoarchitecture of the DG (double-headed arrows denote the dispersion of the GCL) and morphological features of individual eGFP-labeled granule cells, respectively (left panel: Overview; upper and lower right: High-magnification of granule cell dendrites or somata, respectively; red arrowheads, axon initial segments; red arrows, stem dendrites; red open arrows, dendritic swellings; red asterisks, degenerating granule cells. (**B**) Quantification of the GCL width ($n_{cntrl}$ = 7, $n_{1d}$ = 3, $n_{4d}$ = 5, $n_{7d}$ = 6, $n_{14d}$ = 3, $n_{21d}$ = 6) and (**D**) the diameter of initial axons and stem dendrites ($n_{cntrl}$ = 5, $n_{1d}$ = 3, $n_{4d}$ = 5, $n_{7d}$ = 4, $n_{14d}$ = 3, $n_{21d}$ = 5). (**E1-6**) Representative confocal z-plane images of ZnT-3 staining in the DG to determine the dynamics of mossy fiber sprouting (red arrows; $n_{cntrl}$ = 8, $n_{1d}$ = 4, $n_{4d}$ = 5, $n_{7d}$ = 5, $n_{14d}$ = 5, $n_{21d}$ = 9). Locations of granule cell somata are spared (dotted outlines). (**F**) Quantitative analysis of ZnT-3 optical density. (**G1-6**) Representative confocal images of GFAP-stained sections (upper panel) and corresponding 3D-reconstruction in the GCL (lower panel). h, hilus; gcl, granule cell layer; ml, molecular layer; slm, stratum lacunosum moleculare. (**H–I**) Quantitative analysis of the mean and total volume of GFAP-stained processes from radial glia cells in the GCL ($n_{cntrl}$ = 5, $n_{1d}$ = 4, $n_{4d}$ = 5, $n_{7d}$ = 4, $n_{14d}$ = 4, $n_{21d}$ = 5). All statistics were performed with one-way ANOVA, Dunnett's post-test (compared to saline controls); *p<0.05, **p<0.01, ***p<0.001. (**J, K**) Representative sections stained for DAPI (turquoise) and GFAP (magenta) in controls and chronic epileptic mice (21d following kainate injection), respectively. Dashed lines denote the borders of the GCL. (**L**) Quantitative analysis for the optical density of GFAP in individual sections from three controls and in sections from three kainate-injected mice exhibiting weak and strong GCD (one-way ANOVA, Dunnett's post-test; *p<0.05, **p<0.01, ***p<0.001; number of sections, $n_{cntrl}$ = 51, $n_{w-GCD}$ = 6, $n_{s-GCD}$ = 14). (**M**) Corresponding linear regression analysis for the integrated GFAP optical density and (**I**) the area of the GCL (Pearson's correlation). Scale bars in A, 50 μm; in C (left), 100 μm; in C (right), 10 μm; in E (left), 100 μm; in E (right), 10 μm; in G, 30 μm; in K, 200 μm.

## Microstructural alterations during intermediate epileptogenesis correspond to disease progression

Next, we monitored the dynamics of microstructural alterations in vivo using DWI, in order to test whether they could predict the later extent of histopathological changes (*Figure 5*).

In fact, DWI reflected the extensive changes of hippocampal microstructure during epileptogenesis (*Figure 5A,B*), characterized by a robust increase of dorsoventrally-oriented streamlines in the GCL and a decrease in the stratum radiatum of CA1 (*Figure 5E,G*). Correspondingly, fractional anisotropy (as a measure of tissue organization) was increased in the DG during late epileptogenesis (*Figure 5O*; *16d*: 19.9%, p<0.05; *31d*: 28.1%, p<0.001), but exhibited an early and transient decrease in the CA1 region (*Figure 5—figure supplement 1G*; *1d*: −17.8%, p<0.05). Direct histological analysis suggested that these changes indeed correspond to GCD and the loss of CA1 pyramidal cell dendrites, respectively (*Figure 5F,H*). Detailed quantitative analysis revealed that all DWI metrics (mean diffusivity, MD; axial diffusivity, AD; radial diffusivity, RD; fractional anisotropy, FA) progressively increased in the DG and exhibited similar dynamics during epileptogenesis (*Figure 5I, K,M,O*). Particularly, the rise of AD during intermediate disease progression showed the most reliable correlation with chronic histopathological changes: AD significantly increased as early as 8 days after SE in epileptic mice (*Figure 5M*; *8d*: 10.6%, p<0.01; *16d*: 19.9%; *31d*: 18.0%, both p<0.001) and was correlated with the volume of the GCL (*Figure 5N*; *8d*: $r^2$ = 0.68, p<0.01; *16d*: $r^2$ = 0.84, p<0.001; *31d*: $r^2$ = 0.66, p<0.01), suggesting that AD is sensitive to microstructural alterations associated with GCD. No significant diffusivity changes were found in the piriform cortex or the amygdala (*Figure 5—figure supplement 2*).

Given that radial glia cells appear to enlarge and proliferate, concomitantly with the progression of GCD (*Figure 5G–M*), it is conceivable that dorsoventrally-directed radial gliosis might represent an early anatomical substrate affecting the DWI metrics, in particular diffusivity along the dorsoventral axis (dvD) in the DG (*Figure 6*). Indeed, dvD increased continuously reaching significance at day 8 (*Figure 6F*; *8d*: 22.7%; *16d*: 26.6%; *31d*: 21.7%, all p<0.001) similar to the rise in AD. Importantly, from 4 days onward the magnitude of dvD as well as its calculated volume was highly correlated with the later GCL volume (*Figure 6G*) and the integrated density of GFAP labeling in the GCL (*Figure 6J*; *4d*: $r^2$ = 0.61, p<0.01; *8d*: $r^2$ = 0.85; *16d*: $r^2$ = 0.87; *31d*: $r^2$ = 0.81; all p<0.001). This suggests that radial gliosis largely affects early diffusivity changes. However, mossy fiber sprouting and reorganization of granule cells (*Figure 4A–F*) likely contribute to changes of DWI metrics observed during further disease progression.

In summary, our results show that during epileptogenesis microstructural reorganization in the DG identifies disease progression and predicts the later severity of GCD associated with HS.

## Validation of DWI biomarkers in the human dentate gyrus

Next, we investigated the potential translational value of the observed microstructural changes as an imaging biomarker in humans based on ex vivo DWI scans and subsequent histology on resected hippocampi from seven patients with intractable mTLE (*Figure 7*). Quantitative DWI evaluation, however, was only feasible in five hippocampi due to limited tissue integrity and slice thickness.

Similar to our results obtained in mice, human epileptic hippocampi with strong HS (i.e. Wyler grade III or IV) exhibited elevated MD and FA values in the DG, while FA was reduced in the CA1 region (*Figure 7H–K*). These changes could still be detected at lower spatial resolutions achievable in a clinical setting (*Figure 7—figure supplement 1*). Moreover, quantitative morphometry revealed that the density and volume of GFAP-labeled radial glia cell processes also increased with the Wyler grade (*Figure 7L,N*) demonstrating that radial gliosis accompanies HS also in humans. Conversely, only the volume but not the density of Synaptoporin-labeled mossy fiber boutons appear to increase with the Wyler grade (*Figure 7M,O*). Importantly, FA values in the GCL were positively correlated with the density of radial glia processes (*Figure 7P*; $r^2$ = 0.78, p<0.05), suggesting that DWI biomarkers sensitive to microstructural changes in the DG are applicable in intractable mTLE in humans.

## PCA-based evaluation of imaging biomarkers

In our MRI dataset several biomarkers were inter-correlated (*Figure 8A*), presumably because they originate from similar physiological processes and/or are mathematically dependent. In order to clarify which biomarkers are interchangeable, and which are more suitable to predict the severity of

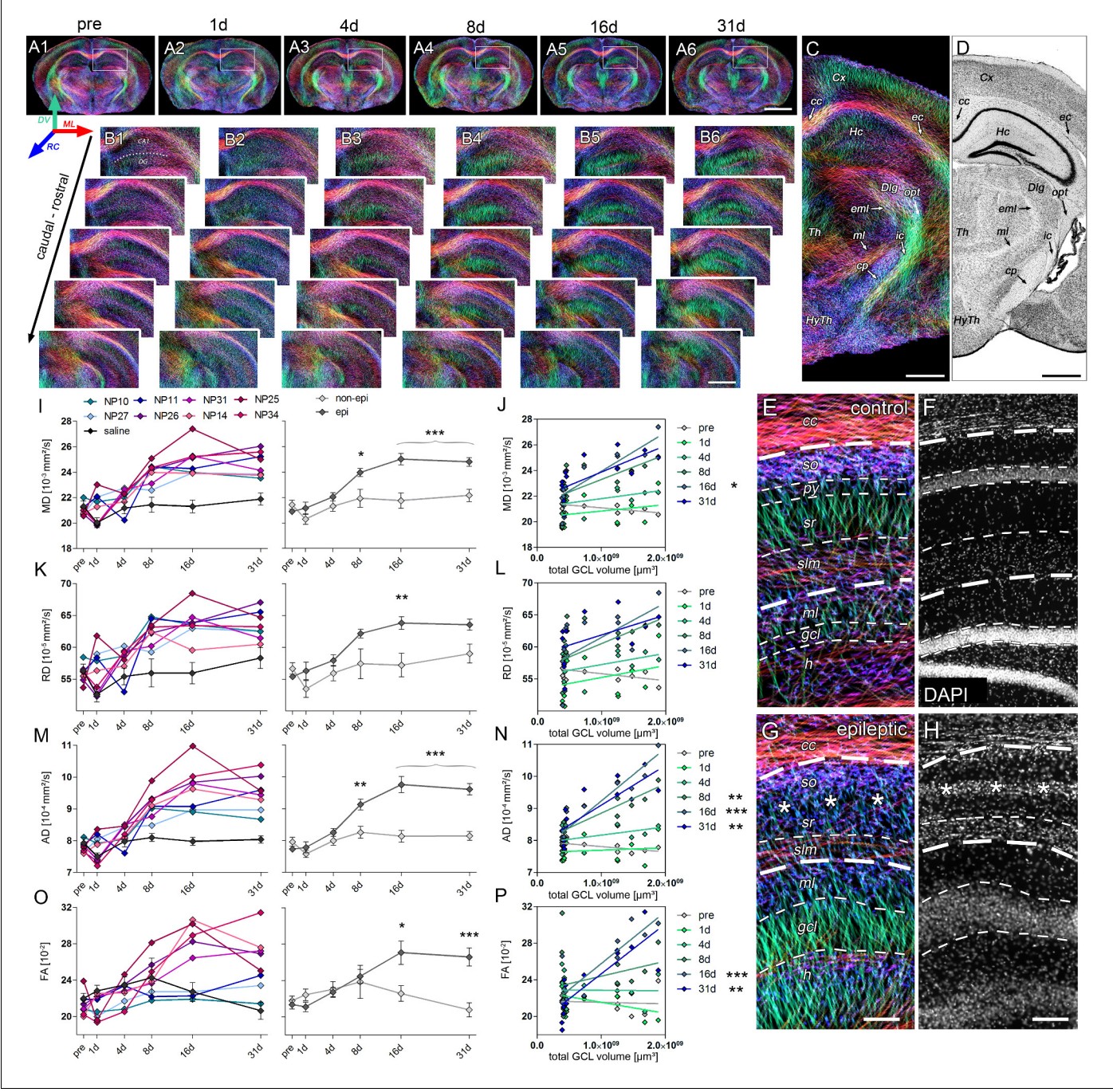

**Figure 5.** Microstructural reorganization quantified by DWI during epileptogenesis predicts disease progression. (**A1-6**) Representative coronal sections from diffusion-weighted tractography at different time points during epileptogenesis (before injection = pre; 1d, 4d, 7d, 14d and 31d following SE). (**B1-6**) Enlarged images of the ipsilateral hippocampus at different levels along the rostro-caudal axis. The orientation of computed streamlines is color-coded [dorsoventral (DV), turquoise; mediolateral (ML), red; rostrocaudal (RC), blue]. (**C–D**) Representative tractography image and a Nissl-stained section (modified from Paxinos and Franklin, The Mouse Brain in Stereotaxic Coordinates, 2001) of corresponding brain regions for anatomical comparison. Computed fibers relate to major axonal pathways and brain regions exhibiting highly oriented dendrites (cc, corpus callosum; cp, cerebral peduncle; Cx, cortex; Dlg, dorsal lateral geniculate nucleus; ec, external capsule; eml, external medullary lamina; Hc, hippocampus; HyTh, hypothalamus; ic, internal capsule; ml, medial lemniscus; opt, optic nerve; Th, thalamus). (**E, G**) Enlarged tractography images demonstrating the distinct orientation of streamlines in different hippocampal layer (dashed lines; cc, corpus callosum; so, stratum oriens; py, pyramidal layer; sr, stratum radiatum; slm, stratum lacunosum moleculare; ml, molecular layer; gcl, granule cell layer; asterisks denote the region of pyramidal cell loss). (**F, H**) Corresponding DAPI-stained sections. Scale bars in A, 2 mm; B-D, 500 μm; H (left), 100 μm. (**I, K, M, O**) Quantitative analysis of DWI metrics [mean- (MD), radial (RD), axial diffusivity (AD) and fractional anisotropy (FA)] in the DG, plotted for individual mice (left panel; controls, black, n = 5; kainate-

*Figure 5 continued on next page*

*Figure 5 continued*

injected animals color-coded) and for groups (right panel; epileptic, dark grey; non-epileptic, light grey) during epileptogenesis. Source data is provided in '*Figure 5—source data 1*'. Two-way ANOVA, Bonferroni's post-test, **p<0.01; ***p<0.001; $n_{non-epi}$ = 6, $n_{epi}$ = 7. Values are presented as the mean ± SEM. (J, L, N, P) Corresponding linear regression analysis of DWI metrics at distinct time points during epileptogenesis with the total GCL volume in epileptic and non-epileptic mice (n = 13; Pearson's correlation, corrected for multiple comparison; *p<0.05, **p<0.01, ***p<0.001). Refer to '*Figure 5—figure supplement 1*' for DWI metrics acquired in CA1.

The following source data and figure supplements are available for figure 5:

**Source data 1.** Summary of DWI metrics.

**Figure supplement 1.** DWI changes in CA1 during epileptogenesis are poor predictors for hippocampal sclerosis.

**Figure supplement 2.** DWI changes in the piriform cortex and the amygdala during epileptogenesis.

---

histopathological changes, we performed a principal component analysis (PCA). Mapping principal components (PCs) of the biomarker dataset onto histopathological changes associated with HS, we found a strong linear relationship between PC1 and GCD ($r^2$ = 0.86, p<0.01), microgliosis ($r^2$ = 0.68, p<0.05) and a trend for radial gliosis (*Figure 8B–D*). Higher order PCs were not significantly correlated with histopathology. Control animals formed a well separated group on the PC1-axis, indicating that PC1 is not only suitable to predict the histopathological changes in the epileptic group, but also to distinguish epileptic from healthy animals. The first two PCs accounted for 58% of the variation (PC1, 38%; PC2, 20%). To estimate how much information the individual markers share with the first two PCs, we correlated each biomarker with PC1 and PC2 and identified four distinct clusters (*Figure 8E*). The largest cluster was especially interesting due to its strong correlation with PC1 ($r^2$ = 0.73, black variables). In fact, a subset of biomarkers of this cluster was remarkably close to PC1 = 1, comprising early changes ($T_2$ in CA1 and [1]H-MR for NAA, glutamate and GABA), but also intermediate changes (DWI metrics) during epileptogenesis. Early alterations in diffusivity grouped into a separate cluster (white variables) and were represented almost exclusively by PC2. Conversely, changes in myoinositol, early and late lactate or FA, and intermediate GABA (light grey variables) were represented by PC1 (negative correlation) and PC2 (positive correlation). Similarly, intermediate NAA, glutamate and FA as well as early-to-intermediate changes of $T_2$ intensity in the DG were correlated with PC1 and PC2 (dark grey variables; positive correlation with both PCs).

Finally, we tested whether our PCA-validated MRI biomarkers also relate to the seizure burden in chronic stages of mTLE. Therefore, we identified seizure-like episodes (*Figure 9A*; spike-and-wave trains >10s) in intra-hippocampal EEG recordings of epileptic mice (n = 6) using an automated algorithm. Across animals we found a considerable variability with respect to their individual seizure frequency (*Figure 9B*; NP11 = 0.73 ± 0.09; NP14 = 0.47 ± 0.11; NP25 = 0.30 ± 0.05; NP27 = 0.73 ± 0.01; NP31 = 0.64 ± 0.10; NP34 = 0.45 ± 0.04 seizure-like episodes / min). Correlating the seizure frequency with PC1 revealed an inverse relationship between both parameters (*Figure 9*; $r^2$ = 0.94, p<0.01), suggesting that our identified biomarkers are also predictive for the seizure severity in chronically epileptic mice.

In summary, our PCA results corroborate the idea that $T_2$-weighted imaging and [1]H-MR spectroscopy are useful tools for early prognosis (1–4 days after the precipitating insult) of later epilepsy severity, whereas DWI is informative at subsequent stages of disease development.

## Discussion

The present study provides a comprehensive, time-resolved analysis of MRI biomarkers for pharmacoresistant mTLE. The identification of biomarkers and their prognostic value was based on the variable severity of histopathological and pathophysiological changes among intrahippocampally kainate-injected mice. Importantly, this well-regarded mTLE model replicates all the major pathological hallmarks of the human disease (*Bouilleret et al., 1999*; *Lévesque and Avoli, 2013*), comprising SE-induced acute unilateral hippocampal cell loss, progression of HS and network reorganization, as well as the emergence of spontaneous recurrent seizures in the chronic epileptic stage. Although

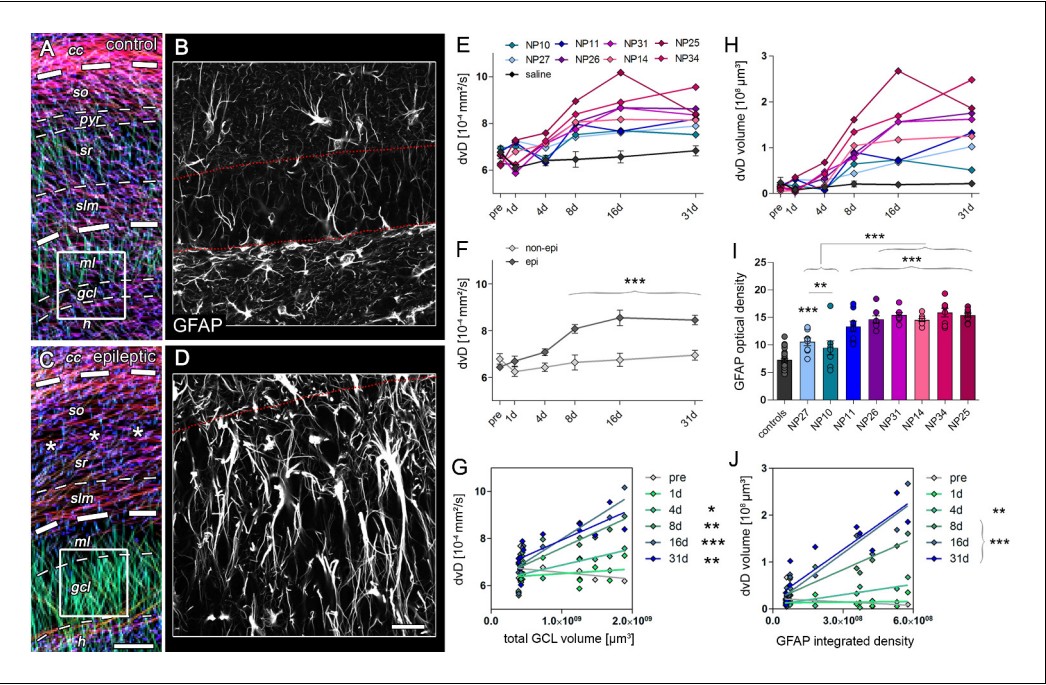

**Figure 6.** Radial gliosis contributes to DWI metrics. (**A, C**) Representative DWI tractography maps of control and epileptic mice. Dashed lines, borders of the hippocampal layers; cc, corpus callosum; so, stratum oriens; py, pyramidal layer; sr, stratum radiatum; slm, stratum lacunosum moleculare; ml, molecular layer; gcl, granule cell layer. Scale bar, 100 μm. (**B, D**) High-magnification confocal images of GFAP staining in the region-of-interest corresponding to A and C (indicated as white boxes). Scale bar, 30 μm. (**E, H**) Quantitative analysis of dorsoventral diffusivity (dvD) in the DG and the calculated volume of increased dvD, respectively, plotted for individual mice (color-coded). (**F**) Groups analysis of dvD for epileptic (dark grey) and non-epileptic (light grey) during epileptogenesis. Source data is provided in '*Figure 6—source data 1*'. Two-way ANOVA, Bonferroni's post-test, **$p < 0.01$; ***$p < 0.001$; $n_{non-epi} = 6$, $n_{epi} = 7$. Values are presented as the mean ± SEM. (**I**) Quantitative analysis of GFAP optical density plotted for controls (black, n = 5) and individual kainate-injected mice (color-coded). One-way ANOVA, Bonferroni's post-test, **$p < 0.01$; ***$p < 0.001$, number of sections, $n_{cntr\ l} = 40$ (8 sections from five controls each), $n_{NP10} = 8$, $n_{NP11} = 8$, $n_{NP14} = 8$, $n_{NP25} = 8$, $n_{NP26} = 8$, $n_{NP27} = 8$, $n_{NP31} = 8$, $n_{NP34} = 8$. (**G, J**) Linear regression analysis of dvD and the calculated dvD volume at distinct time points during epileptogenesis with the total GCL volume and the integrated density of GFAP staining, respectively, in epileptic and non-epileptic mice (n = 13; Pearson's correlation, corrected for multiple comparison; *$p < 0.05$, **$p < 0.01$, ***$p < 0.001$).

The following source data is available for figure 6:

**Source data 1.** Summary of dorsoventral diffusivity metrics.

the initial epileptogenic insult triggering mTLE in humans is apparently different (*Lukasiuk and Becker, 2014*), HS is the common pathology in both human mTLE and animal models (*Wieser and ILAE Commission on Neurosurgery of Epilepsy, 2004*; *Thom, 2014*), and is thought to be critically involved in seizure generation (*Pallud et al., 2011*; *Krook-Magnuson et al., 2015*; *Walker, 2015*). Similar to the human pathology, we found inter-individual differences in the disease severity of epileptic mice, determined by the degree of pyramidal cell death-associated microgliosis and GCD, as well as the frequency of paroxysmal episodes in chronic epilepsy. In a parallel set of experiments we also show that these seizure-like events typically occur in a highly recurrent fashion around the second week after SE, which is in line with previous observations in the same model (*Riban et al., 2002*; *Arabadzisz et al., 2005*; *Heinrich et al., 2011*). Thus, we consider the emergence of recurrent paroxysmal episodes at around two weeks after SE as the onset of the chronic phase. Remarkably, during the first two weeks – typically considered as the latent phase - the dynamics of DWI metrics in the hippocampus, its histopathological reorganization and the development of epileptic activity appeared similar. We would like to emphasize, however, that it was unfeasible to directly relate MRI

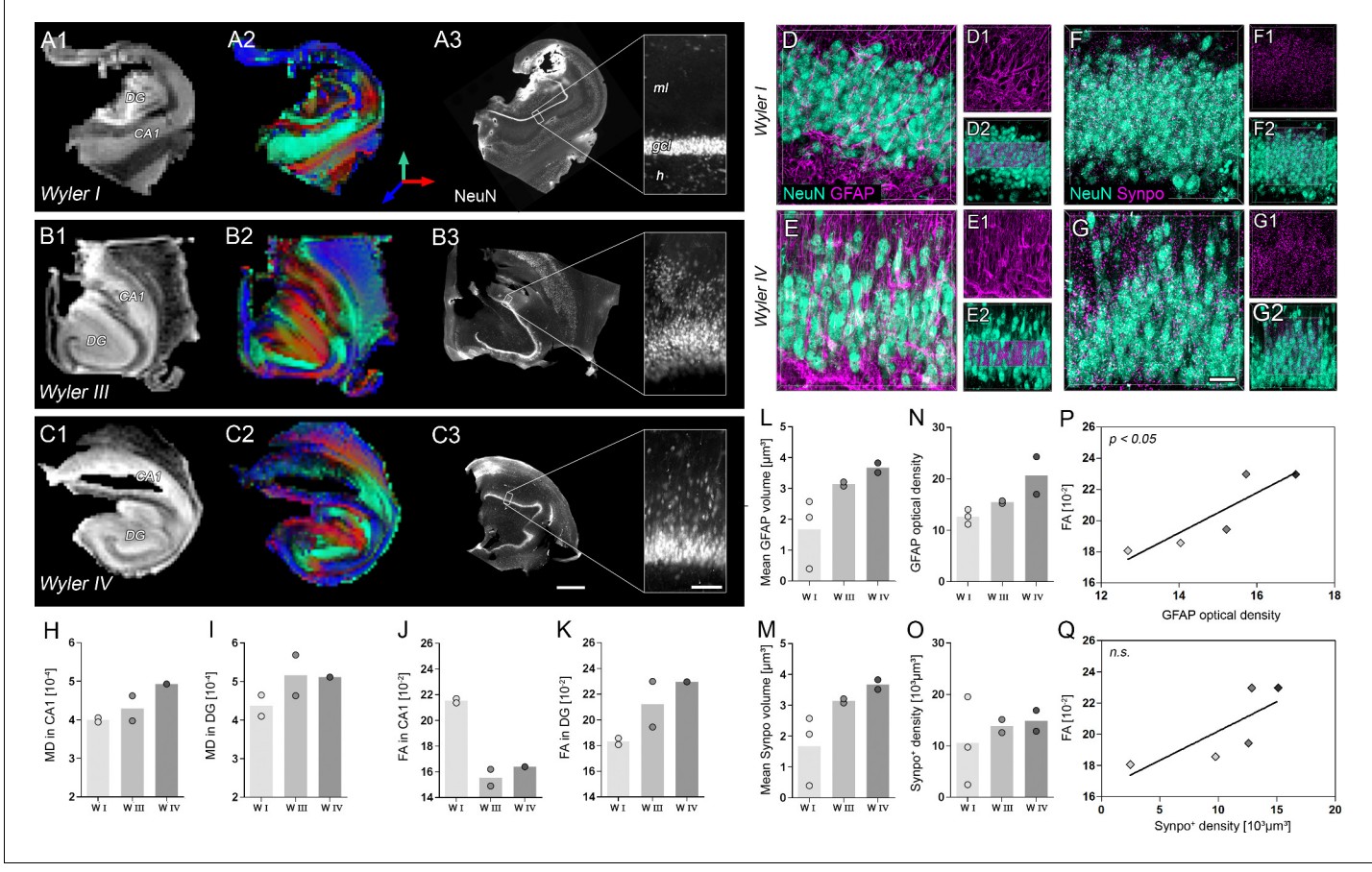

**Figure 7.** Validation of DWI biomarkers in human mTLE. (**A1–C1**) $T_2$-weighted images of sclerotic human hippocampi from mTLE patients scanned ex vivo. (**A2–C2**) Corresponding FA maps (color-coded for orientation: dorsoventral, turquoise; mediolateral, red; rostrocaudal: blue). Mean diffusivity (MD) and fractional anisotropy (FA) values for CA1 and DG are denoted in the lower-left, respectively. (**A3–C3**) Representative NeuN staining of scanned sections. Different Wyler grades according to the severity of neuronal loss (W I: mild, W III: moderate; W IV: strong). Scale bars, 3 mm; inset, 200 μm. (**D–E**) Representative confocal images of hippocampal sections (Wyler I and IV) double immunolabeled for NeuN (turquoise, neurons) and GFAP (magenta, astrocytes and radial glia cells) or (**F–G**) NeuN and Synaptoporin (magenta, mossy fiber synapses), respectively, revealing differences in the microstructure of the human DG between Wyler grades. (**D1–E1, F1–G1**) GFAP and Synaptoporin staining alone. (**D2–E2, F1–G2**) NeuN staining displayed together with reconstructed radial glia processes and mossy fiber boutons within the GCL, respectively. (**H–I, J–K**) Quantitative analysis for MD and FA in CA1 and in the DG, respectively ($n_{WI} = 2$, $n_{WIII} = 2$, $n_{WIV} = 1$; no statistical test performed). Refer to '*Figure 7—figure supplement 1*' for MRI metrics acquired lower scanning resolution. (**L**) Quantitative analysis for the mean volume of GFAP-labeled radial glia processes as well as (**M**) Synaptoporin-labeled mossy fiber synapses, (**N**) and for GFAP optical density as well as (**O**) the density of Synaptoporin-labeled profiles within the GCL ($n_{WI} = 3$, $n_{WIII} = 2$, $n_{WIV} = 2$; no statistical test performed). (**P, Q**) Correlation of FA values with the GFAP optical density and the density of Synaptoporin-labeled profiles, respectively ($n_{WI} = 2$, $n_{WIII} = 2$, $n_{WIV} = 1$; Pearson's correlation).

The following figure supplements are available for figure 7:

**Figure supplement 1.** Comparison of high- and low-resolution ex vivo DWI.

**Figure supplement 2.** Imaris-based 3D-reconstruction.

changes to the progression of epileptic activity, since chronically implanted electrodes would have massively interfered with MRI measurements.

Probing our hypothesis that initial SE-induced hippocampal damage might predict later histopathological severity, we performed longitudinal high-resolution $T_2$-weighted imaging and $^1$H-MR spectroscopy at 7T using a mouse brain adapted cryo-coil. We found an acute and transient increase of hippocampal $T_2$ intensity. Accordingly, previous studies have shown that pyramidal cells and hilar neurons are massively lost 1 day following SE (*Heinrich et al., 2006*; *Marx et al., 2013*),

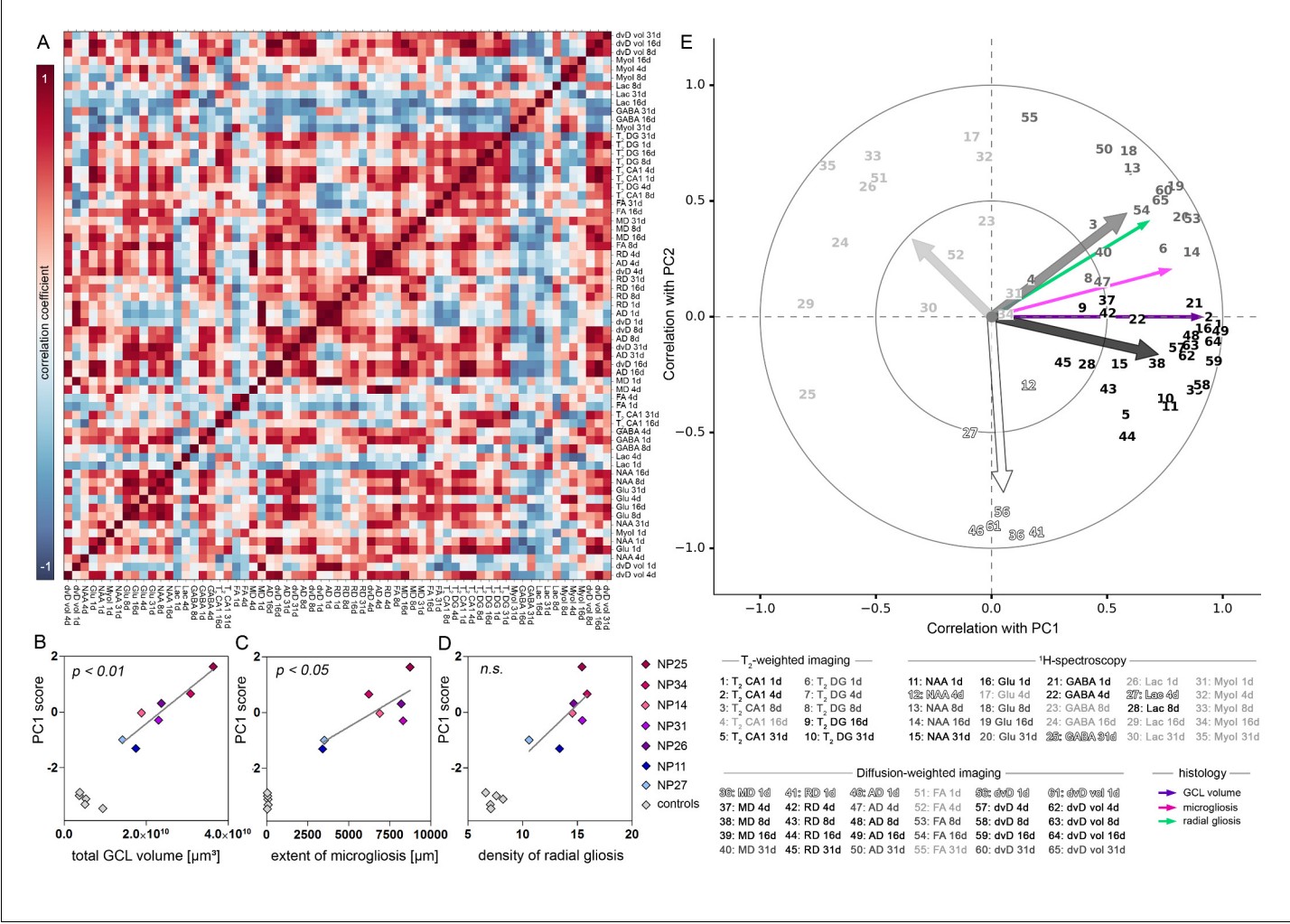

**Figure 8.** PCA-based evaluation of MRI biomarkers. (**A**) Correlation matrix of imaging parameters used for PCA. Only data of kainate-injected mice was analyzed. Positive and negative correlation coefficients are color-coded in red and blue, respectively. (**B**) Pearson's correlation analysis of PC1 scores and total GCL volume, (**C**) extent of microgliosis and (**D**) radial gliosis (i.e optical density of GFAP in the GCL) in epileptic mice (color-coded, n = 7). Controls (grey, n = 5) are plotted but not included in the analysis. (**E**) Correlation plot showing the similarity of individual MRI variables with PC1 and 2. Clusters of variables, indicated as numbers, are grey scale-coded. Arrows denote the population vector of the corresponding cluster. Coefficients are indicated as circles (small = 0.5; large = 1.0).

demonstrating the spatial and temporal specificity of the observed $T_2$ hyperintensity with respect to cell death. In line with our interpretation, increased $T_2$ intensity is thought to represent acute cell loss, subsequent edema, and gliosis (*Jackson et al., 1993*; *Wall et al., 2000*). An acute rise of hippocampal $T_2$ intensity had already been demonstrated at 4.7T by *Bouilleret et al. (2000)* following intrahippocampal kainate application, however, the authors did not assess their prognostic value with respect to disease severity. Our present analysis reveals that this signal increase predicts the degree of later HS. Similarly, early $T_2$ changes were also found in the hippocampus of pilocarpine-injected rats predicting its volume reduction in chronic stages of epilepsy (*Choy et al., 2010*). In contrast, pilocarpine-injected mice exhibited increased hippocampal $T_2$ values after the first week of epileptogenesis, indicating that the extent and time course of SE-induced hippocampal damage also depends on the disease model (*Kharatishvili et al., 2014*). Beyond animal studies, transient $T_2$ hyperintensity was reported in children 3 or 5 days after experiencing prolonged febrile seizures (*Scott et al., 2002, 2003*; *Shinnar et al., 2012*). Importantly, this rise of $T_2$ intensity predicted the

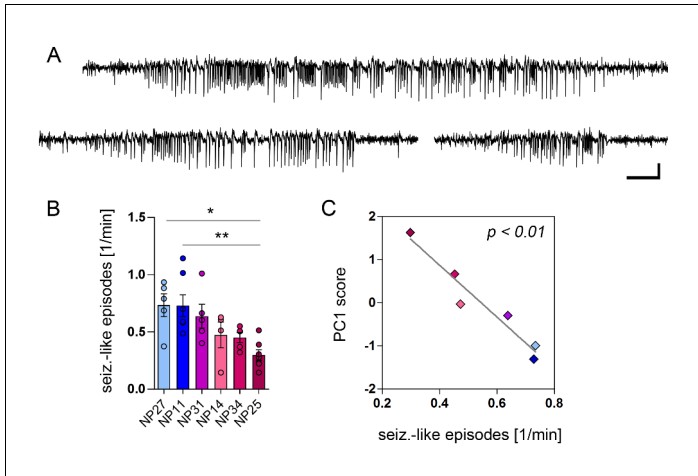

**Figure 9.** PCA-verified biomarkers predict the seizure severity. (**A**) Representative EEG recordings from the ipsilateral hippocampus showing three examples of identified seizure-like episodes. Scale bars: Horizontal 5 s, vertical 2 mV. (**B**) Quantitative analysis of seizure-like episodes for individual epileptic mice (color-coded; note that EEG data is lacking for NP26). One-way ANOVA, Bonferroni's post-test, *p<0.05, **p<0.01, number of recordings: $n_{NP27} = 5$, $n_{NP11} = 7$, $n_{NP31} = 5$, $n_{NP14} = 4$, $n_{NP34} = 5$, $n_{NP25} = 7$. Values are presented as the mean ± SEM. (**C**) Pearson's correlation analysis of PC1 scores and the frequency of seizure-like episodes (n = 6).

The following figure supplement is available for figure 9:

**Figure supplement 1.** Experimental design to translate MRI biomarkers into the clinic.

emergence of HS in longitudinal studies, but its relation to epilepsy onset is yet unclear (*Provenzale et al., 2008*; *Lewis et al., 2014*).

Given that $T_2$ hyperintensity is not inevitably linked to cell loss (*Dubé et al., 2004*; *Thom et al., 2005*), [1]H-MR spectroscopy represents a practical complementary method to monitor tissue damage. Reduced NAA levels can successfully identify the sclerotic hippocampus in both human epilepsy patients (*Ng et al., 1994*; *Hetherington et al., 2002*) and animal models (*Tokumitsu et al., 1997*; *Gomes et al., 2007*; *Filibian et al., 2012*), which is in line with our results showing an early and long-lasting decrease of NAA in the epileptogenic hippocampus. Moreover, we show that, as early as 1 day following SE, reduced NAA correlated with the extent of cell loss-associated microgliosis. In addition, we quantified alterations in the concentration of the major neurotransmitters synthesized in excitatory principal cells (glutamate) and inhibitory interneurons (GABA). Both neuron populations are known to be substantially diminished in the sclerotic hippocampus (*Marx et al., 2013*; *Thom, 2014*). Consistent with these findings, we found that glutamate and GABA rapidly decrease following SE. However, one has to bear in mind that during and early after SE dramatic changes occur in the hippocampus, including cerebral edema and compensatory changes that affect [1]H-MR metrics at early time points after injury. Similar to NAA, the early decrease of glutamate levels was long-lasting and predicted the degree of HS. Conversely, GABA levels increased again at 8 days onwards. Taking into account that hippocampal neurodegeneration is thought to be progressive in rodent mTLE (*Pitkänen et al., 2002*; *Nairismägi et al., 2004*); and own unpublished observations and in human patients (*Kälviäinen et al., 1998*; *Briellmann et al., 2002a*), both long-lasting glutamate decline as well as the restoration of GABA levels following SE appear counterintuitive. We hypothesize that during epileptogenesis the sprouting of glutamate and GABA-producing mossy fibers (own unpublished observations) likely obscures ongoing neurodegeneration on the molecular level.

Accordingly, microstructural reorganization subsequent to hippocampal cell loss was monitored by DWI determining the speed and orientation of water diffusion along biological membranes (*Mori and Zhang, 2006*) for example of axonal fibers (*Harsan et al., 2010*, *2013*). In the epileptic hippocampus of several rodent epilepsy models DWI successfully identified changes of water

diffusion, which is discussed to primarily relate to mossy fiber sprouting and reorganization of myelinated axons (*Kharatishvili et al., 2007*; *Kuo et al., 2008*; *Laitinen et al., 2010*; *Sierra et al., 2015a*, *2015b*; *Salo et al., 2017*). However, it is important to note that the above mentioned epilepsy models do not fully reflect the histopathological hallmarks of pharmacoresistant mTLE (i.e., unilateral HS including GCD) complicating a translation to the human disease. Using the intrahippocampal kainate mouse model we demonstrate that the degree of HS-associated microstructural alterations can be predicted by the progressive increase of diffusivity in the DG as early as 4 days following SE. Similar results were obtained in epileptic rodents, either after traumatic brain injury or after systemic pilocarpine injection (*Kharatishvili et al., 2007*, *2014*), suggesting that DWI biomarkers identified in our study are robust among different models. In rats undergoing traumatic brain injury the increase of hippocampal mean diffusion spans approximately three months (*Kharatishvili et al., 2007*), indicating a longer duration of disease progression in this model. Conversely using pilocarpine-injected mice, *Kharatishvili et al. (2014)* showed that the apparent diffusion coefficient in the hippocampus increased within the first week following SE, predicting the spiking frequency, however, the relation to seizure frequency remained unclear. Moreover, several studies using systemic epilepsy models also revealed $T_2$ or diffusivity changes in the piriform cortex or the amygdala (*Roch et al., 2002*; *Choy et al., 2010*, *2014*; *Kharatishvili et al., 2014*), suggesting that in these animal models both, the hippocampus as well as extrahippocampal brain regions sensitive to excitotoxic injury are affected. We did not observe these changes, highlighting the key role of the lesioned hippocampus in driving epileptogenesis in the intrahippocampal kainate mouse model. As discussed by *Gomes and Shinnar (2011)*, an attractive biomarker should ideally infer the progression of epileptogenesis and not only its presence. Indeed, DWI changes in the DG steadily progressed over the first 2 weeks of epileptogenesis, resembling the aggravation of epileptiform activity during this period (*Heinrich et al., 2011*). Importantly, the fact that early MRI biomarkers are strongly correlated to later biomarkers makes epileptogenesis as a whole predictable. Early changes (i.e. cell death identified by $T_2$ and $^1$H-MR) might scale more or less linearly to the later disease development (i.e., microstructural reorganization monitored by DWI), however, it remains to be determined whether different MRI dynamics might also reflect different times of seizure onset. Given this interconnectivity of MRI biomarkers, our PCA results validate early changes in $T_2$, $^1$H-MR and subsequent changes in DWI metrics as highly prognostic for both, later HS severity and seizure frequency. Interestingly, PC1 is positively correlated with histopathological changes associated with HS, but negatively with seizure frequency. This data indicates that more severe early hippocampal injury and subsequent tissue reorganization lead to the development of stronger HS and lower seizure burden, compared to a milder, but still epileptogenic insult. We hypothesize that hippocampal damage needs to exceed a certain threshold until epilepsy develops, whereas further cell loss might attenuate epilepsy severity because fewer neurons are able to participate in seizure generation.

Investigating the anatomical correlates of DWI changes further, our in-depth morphological analysis clearly suggests that the early (4 to 8 days) rise of DWI values is largely driven by radial gliosis, whereas in later stages (8 days onward) neuronal hypertrophy, mossy fiber sprouting and translocation of granule cell dendrites might also contribute. Our results are in line with *Budde et al. (2011)* and *Salo et al. (2017)* showing that FA increases with glial scarring in the perilesional cortex following cerebral trauma or the CA3 region following SE, respectively. The notion that changes in DWI metrics are influenced by gliosis is further supported by our observation that DWI changes in the DG were accompanied by $T_2$ hyperintensity, known to correlate best with the number of local glia cells (*Briellmann et al., 2002b*). Conversely, in systemic epilepsy models reorganization of myelinated fibers in the molecular layer, but not of local astrocytes, appear to effectively drive DWI changes in the DG, particularly with respect to FA and AD (*Salo et al., 2017*). Considering that in these models GCD is not present, we conclude that the region-specific orientation of glial cells in the DG and their reorganization during GCD formation is critical to affect DWI metrics. Accordingly, as a prerequisite for radial glia cells to affect diffusion anisotropy, they traverse the GCL perpendicularly and respond quickly to epileptiform activity with hypertrophy, branching and proliferation (*Heinrich et al., 2006*; *Sierra et al., 2015c*). In turn, we show that processes of these glia cells are still dorsoventrally organized in the dispersed septal DG, thus contributing to the observed increase in dvD, AD and FA. Indeed, we demonstrated that FA in the DG of human mTLE patients relate to the increase of radial gliosis at higher Wyler grades. Our finding is supported by previous studies revealing that the radial glia processes traverse the GCL and are correlated with the extent of GCD and HS in patients

(*Fahrner et al., 2007*). Moreover, our analysis demonstrated that MD and FA are inferior to AD and dvD regarding their ability to predict HS-associated microstructural reorganization in the DG. Previous animal studies attributed increased diffusivity and anisotropy to mossy fiber sprouting (*Kharatishvili et al., 2007*; *Parekh et al., 2010*) and reorganization of myelinated fibers (*Laitinen et al., 2010*), but in these studies histological validation was performed only in the chronic stage of epilepsy. Our histological analysis expands this view revealing that significant mossy fiber sprouting is restricted to the later stages of epileptogenesis (around 14 days onward), which is consistent with previous observations for the CA2 region (*Häussler et al., 2016*). Although it is well conceivable that mossy fiber sprouting alters DWI metrics during this period, earlier changes (4 to 14 days) particularly of AD and dvD likely correspond to radial gliosis.

In vivo DWI studies performed in human mTLE consistently reported an increase of MD, but a decrease of FA in the sclerotic hippocampus (*Thivard et al., 2005*; *Salmenpera et al., 2006*; *Liacu et al., 2010*) which can be explained by the expansion of the extracellular space secondary to cell death and the subsequent loss of tissue organization. However, these studies addressed diffusivity changes in the whole hippocampus, disregarding the differential effects of neuronal reorganization in hippocampal subfields (*Janz et al., 2017*). Consistent with this notion, we found an opposing trend of FA values for the CA1 region and the DG in resected hippocampi from mTLE patients with high Wyler grades. Similarly, in epileptic mice we encountered an initial decrease of FA in the CA1 stratum radiatum, but an increase in the GCL, clearly demonstrating the importance of hippocampal subfield imaging with respect to DWI biomarkers. However, one has to bear in mind the high spatial resolution used to detect these subfield-specific changes. Our DWI protocol employed a planar resolution of $60 \times 60$ $\mu m^2$ to image the mouse hippocampus spanning a cross-sectional area of about $2.5 \times 1.5$ $mm^2$. In contrast to our setup optimized for the mouse brain, a clinical setup might reach 800 $\mu m$ isotropic resolution in a 1 hr scan using state-of-the-art imaging sequences at 7T (*Heidemann et al., 2012*). Considering the much larger cross-sectional area of the human hippocampus (about $8 \times 8$ $mm^2$), it should become feasible to retrieve accurate DWI metrics from hippocampal subfields in a clinical setting. Indeed, degeneration of the perforant path in aging humans has already been shown at 3T (*Yassa et al., 2010*). Moreover, imaging of subfield-specific pathology in human TLE was successfully performed at 3T and at 7T at planar resolutions of $1000 \times 1000$ $\mu m^2$ and $500 \times 500$ $\mu m^2$, respectively (*Goubran et al., 2016*). These technical advances provide the foundation for the application of our imaging biomarkers in a clinical setting with the ultimate goal to identify and even prevent epileptogenesis before seizure onset. In this context, longitudinal MRI experiments similar to the FEBSTAT study (*Lewis et al., 2014*) will be essential, to determine the dynamics of hippocampal alterations in humans who just suffered from prolonged seizures (e.g., due to SE, head trauma, febrile seizures or encephalitis). These studies would require repeated MRI measurements over many years, starting with short intervals early after the initial precipitating injury and lasting until the first clinical seizures arise (*Figure 9—figure supplement 1*). MR measurements would need to rely on multiple imaging modalities to integrate the most promising MRI biomarkers evaluated in the present study (i.e., $T_2$ hyperintensity in CA1, changes in hippocampal NAA, glutamate and GABA levels and AD, dvD and FA increase in the DG). We propose that the quantitative relationship between these biomarkers will be highly informative for the course of epileptogenesis and may allow a prognosis for subsequent disease progression and early pharmacological intervention.

## Material and methods

### Animals

Experiments were performed on adult (9–12 weeks) male C57Bl/6N wildtype (Charles River, Sulzfeld, GER) or transgenic Thy1-eGFP mice in which eGFP is expressed under the control of the *Thy1* promoter (RRID:IMSR_JAX:007788, M-line, C57BL/6 background; *Feng et al., 2000*). Each animal represents an individual experiment, performed once. A total of 13 wildtype and 25 transgenic mice were used for longitudinal MRI experiments and complementary histological investigation at individual time points, respectively. Additionally, 8 Thy1-eGFP mice were used for longitudinal EEG recordings. Mice were kept at room temperature (RT) in a 12 hr light/dark cycle providing food and water ad libitum. All animal procedures were in accordance with the guidelines of the European Community's

Council Directive of 22 September 2010 (2010/63/EU) and were approved by the regional council (Regierungspräsidium Freiburg).

## Kainate injections

Mice received a single, unilateral kainate injection into the hippocampus as described previously (*Heinrich et al., 2006*; *Häussler et al., 2012*). In brief, anesthetized mice (ketamine hydrochloride 100 mg/kg, xylazine 5 mg/kg, atropine 0.1 mg/kg body weight, i.p.) were stereotaxically injected with 50 nL of a 20 mM kainate solution (Tocris, Bristol, UK) in 0.9% saline into the right dorsal hippocampus (coordinates relative to bregma: anteroposterior = −2.0 mm, mediolateral = −1.5 mm, and relative to the cortical surface: dorsoventral = −1.8 mm). Controls were injected with 0.9% saline. After recovery from anesthesia, behavioral SE was verified by observing mild convulsions, chewing or rotations. Four mice, which died between the fourth and sixth day following kainate injection, were excluded from the study.

## Human tissue

Seven resected hippocampal specimens from mTLE patients (mean age: 36.8 ± 8.8 years) were used in this study. All patients had experienced pharmacoresistant complex partial seizures and underwent amygdalohippocampectomy to achieve seizure control. Patients were treated according to the Epilepsy Surgery Program of the University Medical Center Freiburg. Informed consent was obtained from all patients. Tissue selection was approved by the Ethics Committee at the University Medical Center Freiburg. Classification of Ammon's horn sclerosis was carried out by a clinical neuropathologist according to Wyler (*Hermann et al., 1992*): Grade 1, slight (<10%) or no loss of pyramidal cells in the cornu ammonis 1–4 (CA1-4); Grade 2, gliosis and moderate loss (10–20%) of CA1, CA3 and/or CA4 pyramidal cells; Grade 3, gliosis with >50% pyramidal cell loss in CA1, CA3 and CA4, but sparing CA2; Grade 4, gliosis with >50% pyramidal cell loss involving CA1-4.

## Magnetic resonance imaging

All MR measurements were performed on a 7T preclinical MRI system equipped with a mouse head adapted $^1$H transmit-receive CryoProbe (MR system: BioSpec 70/20, software ParaVision 6.0, Bruker, Ettlingen, Germany). For animal handling, a dedicated mouse bed (Bruker) was used that provided a three point-fixation system (tooth-bar and ear-plugs), isoflurane anesthesia and stabilization of the body temperature of the mice. Respiration was measured by a pressure sensor and isoflurane anesthesia (1.5%) was adjusted to keep the respiration rate at 50–60 breaths/min during the scans.

At the beginning of each session, a $B_0$ field map (isotropic resolution $0.3 \times 0.3 \times 0.3$ mm$^3$) was acquired and used to improve the $B_0$ field homogeneity inside the mouse brain (calculation of the local map-based shims), followed by adjustments of the local frequency and reference power (flip angle). This process was repeated a second time to optimize the $B_0$ field inside the brain (linewidth of the whole brain water signal 30–50 Hz). For spectroscopy the $B_0$ field was optimized according to the region-of-interest (line width of the unsuppressed water signal inside the voxel 8–12 Hz).

### T$_2$-weighted imaging

A RARE sequence was used with the parameters: TR 3 s, TE 50 ms, RARE factor 4, matrix $440 \times 256$, (interpolated) in-plane resolution $29 \times 29$ μm$^2$, interpolation factor in the read and phase directions 1.4, 22 slices, slice thickness 0.4 mm, NA 6. The acquisition time 828 s was prolonged by respiratory triggering to $\approx 20$ min.

Differences in the distance from the brain to the surface transmit-receive coil induce inhomogeneous acquired signal intensities between scans. To eliminate this effect all images were normalized by equalizing the mean intensity in the ipsilateral thalamus. All acquired images and a labeled mouse brain atlas (Australian Mouse Brain Mapping Consortium AMBMC, University of Queensland, AUS) were registered to one reference data set (*Richards et al., 2011*) using FLIRT (RRID:SCR_002823, FSL toolbox, FMRIB, Oxford, UK). Subsequently mean signal intensities were quantified in hippocampal subfields using MATLAB (MathWorks, Massachusetts, USA).

## $^1$H-MR spectroscopy

$^1$H-MR spectra were acquired from two voxels ($2 \times 1.4 \times 1.4$ mm$^3$) located in the septal part of the ipsi- and contralateral hippocampus, respectively. The parameters of the PRESS sequence were: TR 2.5 s, TE 20 ms, acquisition bandwidth 5000 kHz, 2048 points, VAPOR water suppression, working chemical shift offset $-2.5$ ppm, NA 400, drift and eddy current correction in ParaVision 6.0, acquisition time 1000 s.

The spectra were quantified by LCModel (RRID:SCR_014455, S. Provencher, LCModel Inc., Ontario, CAN) and absolute concentrations were calculated using the unsuppressed water signal as reference. In naive mice, the standard deviations (Cramér-Rao lower bounds) of the relevant compounds were <15% [NAA 3%; glutamate 3%; GABA 8%; myoinositol 4%; lactate 11–14%; full width at half maximum 0.033 ppm and signal-to-noise ratio (SNR) 28].

## Diffusion-weighted imaging

For DWI, a spin-echo EPI sequence was used with the parameters: TR 2.5 s, TE 33 ms, matrix $220 \times 128$, in-plane resolution $58 \times 58$ μm$^2$, interpolation factor in the read and phase directions 1.4, 22 slices, slice thickness 0.4 mm, 3 segments, NA 6, 3 non-diffusion-weighted images, 30 diffusion-weighted images, diffusion gradient duration 2.5 ms, separation 14 ms, b = 1000 s/mm$^2$. The acquisition time of 1485 s was prolonged by respiratory triggering to 45–60 min.

Movements during the scan caused by frequency drifts and actual head motion were corrected by registration of the images to the first non-diffusion weighted image using FLIRT (FSL toolbox). Subsequently the diffusion tensor and corresponding DWI parameter maps (MD, AD, RD, dvD and FA) were calculated. In addition, the extent of the diffusion tensor ellipsoid in the DV direction was calculated as measure for the diffusivity in the DV direction. Registration of images to a reference data set and registration of a labeled brain atlas was done as described in the previous section. Additionally, a microstructural streamline model was reconstructed from the measured high angular resolution diffusion data using a tractography method based on a global optimization approach (*Reisert et al., 2011*). This method globally estimates the optimal structural configuration of connected line segments whose influence on water diffusion best matches the measured MR data.

For ex vivo DWI of human hippocampal tissue, a 3D spin-echo EPI sequence was used with the parameters: TR 4 s, TE 80 ms, 4 segments, matrix $98 \times 98 \times 10$, isotropic resolution $200 \times 200 \times 200$ μm$^3$, 5 non-diffusion weighted images, 50 diffusion directions, 2 b-values: 1000 and 2000 s/mm$^2$, diffusion gradient duration 30 ms, separation 38 ms, NA 1, acquisition time 4 hr and 40 min. Stronger diffusion weighting was applied because the fixation of the tissue (see below) decreases the diffusivity substantially (*Sun et al., 2005*; *Hui et al., 2008*). A longer duration of the diffusion gradients was chosen to achieve gradient conditions that would also be realizable in clinical MRI systems. Visual inspection of the acquired images revealed no frequency drift or motion of the sample during the scan. Accordingly, no motion correction was applied. For quantification of DWI parameters in specific hippocampal subfields, regions-of-interest were manually selected in each sample according to the subfields identified by immunohistochemistry.

## Methodological considerations of MR acquisitions

Some methodological factors might influence the obtained results. The most relevant aspects concern the quantification of the metabolite spectra and the set of diffusion tensor imaging (DTI) acquisition parameters.

There are commonly two different approaches for metabolite quantification, one quantifies metabolite ratios, for example the ratio to total creatine (creatine plus phospho-creatine, CrPCr), and the other quantifies absolute values using water as reference. As SE led to massive cell loss, creatine (and other metabolites) may be altered and not necessarily suitable as a reference. Therefore it was decided to use water for absolute quantification. This may be affected by edema in the tissue, but we found no systematic change of all metabolites that matched the time-course of T$_2$ intensities that could be explained by changes in the water reference. The ratios NAA/CrPCr and glutamate/CrPCr (data not shown) also showed persistent decreases demonstrating that the observed changes were also present independent of the water reference.

In diffusion measurements there is always a tradeoff between desired features – high resolution, strong diffusion weighting (b-value) and many diffusion directions – and the resulting acquisition

time and SNR. For in vivo measurements, acquisition time is critical, necessitating some tradeoffs. This study focused on alterations within the hippocampus, in which much finer structures are present in transverse slices than along the longitudinal axis. Thus, we also utilized an anisotropic voxel geometry. Choosing an isotropic voxel geometry with the same in-plane resolution would drastically reduce the SNR while adding comparatively little information. It has even been shown that in DTI an anisotropic analysis, averaging noise along a rather homogeneous direction and reducing partial volume effects, could lead to increased sensitivity of detecting pathology (*Van Hecke et al., 2010*). For in vivo DTI a b-value of 1000 s/mm2 and 30 diffusion directions is in the optimal range. Although fiber tracking approaches generally benefit from higher b-values und more diffusion directions, higher b-values would have led to low SNR and several studies have validated the results of the applied tractography method using similar acquisition schemes (*Harsan et al., 2013*; *Anastasopoulos et al., 2014*; *Horn et al., 2014*).

## Electrode implantation and local field potential recordings

For EEG recording after MR measurements (31 days after saline or kainate injection), mice were anesthetized and were stereotaxically implanted with platinum-iridium wire electrodes (Ø 125 μm; World Precision Instruments, Berlin, GER) in both hippocampi. Coordinates relative to bregma: anterioposterior = −2.0 mm, mediolateral = +1.4 and −1.4 mm, and relative to the cortical surface: dorsoventral = −1.6 mm. Two stainless-steel screws implanted above the somatosensory cortex were used as reference and ground. Subsequently, mice were connected to a miniature preamplifier (Multi Channel Systems, Reutlingen, GER). Signals were amplified (1000-fold, bandpass 1 Hz to 5 kHz) and digitized (sampling rate 10 kHz; Power1401 analog-to-digital converter; Spike2 software, RRID:SCR_000903, Cambridge Electronic Design, Cambridge, UK). Mice were recorded at five consecutive days, 2–3 hr each day, to check for epileptiform activity. For longitudinal EEG recordings, eight mice were chronically implanted with platinum-iridium wire electrodes into the ipsilateral hippocampus directly after kainate injection and recorded (same parameters as described above) at 6 hr, 1 d, 3–5 d, 6–7 d, 12–16 d and 17–21 d after injection. Epileptic discharges identified by trains of high-amplitude population spikes (≥5 s; see *Figure 1—figure supplement 1E,F*) were quantified by two blinded observers using Spike2. The mean of the values obtained by both observers was used for statistical analysis. Each mouse represents the biological and the number of recordings per mouse the technical replicate. Only recordings from electrodes located in the molecular or granule cell layer (verified in DAPI-stained sections) were used for further evaluation.

### Automated data analysis

Prior to analysis, artifacts were removed by visual inspection of the LFP trace. Epileptiform discharges were detected by a custom algorithm that had been validated by a human observer on a different data set (sensitivity: 0.87, false positives/min: 1.9). Detection performance on our data-set was checked for each recording on individual subsets of hits. We regarded trains of epileptiform discharges as 'seizure-like episodes' when lasting longer than 10 s.

## Tissue preparation and immunohistochemistry

Mice were anesthetized 36 days after kainate injection and transcardially perfused with 0.9% saline followed by 4% paraformaldehyde in 0.1 M phosphate buffer (PB; pH 7.4) for 5 min. Postfixation was performed in the same fixative overnight at 4°C. Brains were subsequently cut (coronal plane, 50 μm) on a vibratome (VT1000S, Leica, Bensheim, GER). Each mouse represents the biological and the number of brain sections per mouse the technical replicate.

Resected human specimens were first trimmed to blocks of about 5 mm thickness and subsequently immersion-fixed in 4% paraformaldehyde in 0.1 M PB overnight at 4°C, transferred to 30% sucrose at 4°C overnight, shock-frozen and finally stored at −80°C. Prior to ex vivo DWI scans, tissue blocks were thawed in 0.1 M PB at RT. For subsequent histology, tissue blocks were further cut into 100 μm serial sections on a VT1000S vibratome (Leica). Each resected hippocampus represents one biological and the number of sections per hippocampus the technical replicate.

For immunofluorescence staining, free-floating serial sections were pre-treated with 0.25% TritonX-100 in 1% bovine serum albumin for 1 hr (mouse) or 2 hr (human). Subsequently, serial tissue sections were incubated with the following primary antibodies at 4°C for 24 hr (mouse) or 48 hr

(human): Guinea pig anti-NeuN (1:500; RRID:AB_2619988, Synaptic Systems, Göttingen, GER), rabbit anti-ZnT-3 (1:2000; RRID:AB_10894885, Synaptic Systems), rabbit anti-Synaptoporin (1:1000; RRID:AB_2619748, Synaptic Systems), rabbit anti-Iba-1 (1:1000; RRID:AB_839504, Wako Chemicals, Neuss, GER) or rabbit anti-GFAP (1:500; RRID:AB_10013482, Dako, Hamburg, GER). For detection, Cy2-, Cy3- or Cy5-conjugated secondary donkey anti-guinea pig or goat anti-rabbit antibodies (1:200; RRID:AB_2340462, RRID:AB_2338000 and RRID:AB_2307385, Jackson ImmunoResearch Laboratories Inc., West Grove, USA) were used. All secondary antibodies were applied for 3 hr (mouse) or 12 hr (human) at room temperature followed by rinsing in 0.1 M PB for $6 \times 15$ min (mouse) or $6 \times 1$ hr (human). Counterstaining was performed with DAPI (4',6-diamidino-2-phenylindole; 1:10.000, Roche Diagnostics GmbH, Mannheim, GER). Sections were mounted on glass slides and coverslipped with ProLong Gold (Molecular Probes, Invitrogen, Carlsbad, USA) the next day.

Fluoro-Jade B (FJB, Millipore, Schwalbach, GER) staining was performed to monitor cell death. Sections were mounted on gelatin-coated glass slides and incubated with 0.06% potassium permanganate solution for 15 min followed by 0.0004% FJB solution for 30 min. Sections were then rinsed in xylene and coverslipped with Hypermount (Thermo Fisher Scientific, Dreieich, GER).

## Histological analysis

To quantify the severity of HS, two parameters (cell loss in CA1-3 and GCD) were investigated using an AxioImager2 microscope (Zeiss, Göttingen, GER). Using a 10x objective (Plan-APOCHROMAT, Zeiss), photomicrograph composites were acquired with a digital camera (MR605, Zeiss) from Iba1- and/or DAPI-stained sections, processed with Zen software (RRID:SCR_013672, Zeiss) and analyzed with Fiji ImageJ software (RRID:SCR_002285, *Schindelin et al., 2012*). The individual extent of pyramidal cell death was estimated by measuring the spatial extent of microgliosis in the cornu ammonis, since activated, amoeboid microglia build a dense scar tightly restricted to the region of pyramidal cell loss (*Figure 1D*). For each mouse eight to twelve Iba-1 stained sections were categorized into five levels (L1-5) according to the rostocaudal axis: Approximate coordinates relative to bregma, L1 = −0.9 to −1.6 mm, L2 = −1.6 to −2.3 mm, L3 = −2.3 to −2.9 mm, L4 = −2.9 to −3.6 mm, L5 = −3.6 to −4.0 mm. The mean length of microglial scarring was calculated for each level and summed up. The degree of GCD, which is characterized by a broadened cell layer and reduced cell density, was assessed by measuring the area of the GCL in DAPI-stained sections (*Figure 1A*) and calculating the overall volume, taking in to account the slice thickness of 50 μm. Since the number of available sections varied across mice (51 ± 7 sections), possibly due to slightly different brain sizes or non-uniform shrinking of perfused tissue, we normalized the rostocaudal axis to 2500 μm (50 sections per animal x 50 μm slice thickness). This ensured that correlations with MRI measures actually represent their relationship to HS-associated anatomical changes rather than the variability in the number of sections. In addition, we estimated individual degree of radial gliosis by quantifying the optical density of GFAP in the GCL of the septal hippocampus (see parameters below) using Fiji ImageJ, followed by multiplying its mean with the calculated GCL volume. Considering that GCD is not present in more temporal regions of the kainate-injected hippocampus, the mean of GFAP control values was used for calculating its integrated density in the non-dispersed GCL.

Microstructural alterations in the DG were analyzed by confocal laser scanning microscopy in MR-scanned wildtype mice and in complementary experiments using transgenic Thy1-eGFP mice. For each animal, four to eight confocal stacks were acquired in two sections with an Olympus FV10i (Olympus Deutschland, Hamburg, GER) at high resolution (60x oil-immersion objective, 1024 × 1024 pixels; 4x frame-average; confocal aperture 1 airy unit; z-step size: 0.5 μm). Laser power and detector sensitivity were kept constant for each staining to allow comparison between samples. For quantitative morphometry, image stacks were transferred to Imaris 7.7.1 software (RRID:SCR_007370, Bitplane AG, Zurich, CH). Morphological changes of eGFP-labeled dentate granule cells (10–20 per z-stack) were investigated by measuring the diameter at three positions along the dendrite and axon segments close to the soma. Changes in the density and volume of GFAP-labeled processes from radial glia cells and Synaptoporin-labeled mossy fiber boutons were inferred by surface reconstruction of a predefined region-of-interest within the GCL (*Figure 7—figure supplement 2*). Additionally, for each sample the optical density of GFAP within the GCL was quantified in six to eight confocal planes taken from the section surface using Fiji ImageJ. Similarly, the amount of sprouted mossy fibers was determined by measuring the optical density of ZnT-3 within the GCL. For optical density measurements the intensity spectrum was optimized for contrast by histogram equalization

to obtain quantitative values representing more the actual density of labeled profiles than differences in labeling intensity itself.

Due to the apparent anatomical changes in the kainate-injected hippocampus blinded observation was unfeasible. However, all analyses were performed semi-automatically, and both MRI and histological data were collected independently and combined subsequently to avoid a putative bias.

## Statistical analysis

Data was tested for statistical significance with Prism five software (RRID:SCR_002798, GraphPad Software Inc., La Jolla, USA). Comparison of two groups was performed with an unpaired Student's t-test (two-tailed). When more than two groups were compared either one-way ANOVA followed by Dunnett's or Bonferroni's post-test, or two-way ANOVA followed by Bonferroni's post-test were used to correct for multiple comparison. Significance thresholds were set to: $*p<0.05$, $**p<0.01$, $***p<0.001$. For all values, mean and standard error of the mean (SEM) are given. Correlations were tested using Pearson's correlation (slope significantly non-zero, confidence interval set to 95%, corrected for multiple comparisons). PCA was performed with Python (RRID:SCR_008394, Python Software Foundation, Beaverton, USA). All statistical results are summarized in *Supplementary file 1*.

## Data availability statement

The MRI dataset has been made available (https://osf.io/7gmvn/).

The source code and user information for the custom seizure detection algorithm can be obtained from Ulrich Egert upon request. Contact: egert@imtek.uni-freiburg.de

## Acknowledgements

We thank Andrea Djie-Maletz and Enya Paschen for excellent technical assistance and Dr. Thomas Lange for useful advice on MRI data analysis. This work was supported by the German Research Foundation as part of the Cluster of Excellence 'BrainLinks-BrainTools' within the framework of the German Excellence Initiative (grant number EXC 1086 to JK, JH, UE, PL, and CAH); by the ERA-Net (grant number NEURON II CIPRESS to CAH); by the Bundesministerium für Bildung und Forschung (grant number FKZ 1GQ0830 to UE), co-financed by the European Union/European Regional Development Fund (UE).

## Additional information

### Funding

| Funder | Grant reference number | Author |
| --- | --- | --- |
| Deutsche Forschungsgemeinschaft | EXC 1086 | JG Korvink<br>Jürgen Hennig<br>Ulrich Egert<br>Pierre LeVan<br>Carola A Haas |
| Bundesministerium für Bildung und Forschung | ERA-Net NEURON II CIPRESS | Carola A Haas |
| Bundesministerium für Bildung und Forschung | FKZ 1GQ0830 | Ulrich Egert |

The funders had no role in study design, data collection and interpretation, or the decision to submit the work for publication.

### Author contributions

PJ, Data curation, Formal analysis, Investigation, Visualization, Methodology, Writing—original draft, Writing—review and editing; NS, Data curation, Formal analysis, Investigation, Visualization, Methodology, Writing—review and editing; KH, Formal analysis, Investigation, Visualization, Methodology, Writing—review and editing; UH, Data curation, Investigation, Methodology, Writing—review and editing; JGK, Supervision, Funding acquisition, Writing—review and editing; DvE, Resources, Methodology; JH, Conceptualization, Resources, Supervision, Funding acquisition; UE, Supervision,

Funding acquisition, Validation, Methodology, Writing—review and editing; PL, CAH, Conceptualization, Supervision, Funding acquisition, Validation, Methodology, Writing—review and editing

#### Author ORCIDs
Ute Häussler, http://orcid.org/0000-0001-5601-9833
Jan G Korvink, http://orcid.org/0000-0003-4354-7295
Ulrich Egert, http://orcid.org/0000-0002-4583-0425
Carola A Haas, http://orcid.org/0000-0002-7022-4136

#### Ethics
Human subjects: Informed consent was obtained from all patients. Tissue selection was approved by the Ethics Committee at the University Medical Center Freiburg.
Animal experimentation: All animal procedures were in accordance with the guidelines of the European Community's Council Directive of 22 September 2010 (2010/63/EU) and were approved by the regional council (Regierungspräsidium Freiburg).

## Additional files

#### Supplementary files
• Supplementary file 1. Quantitative summary of statistically tested parameters. The table displays all results statistical tests performed (right column) for each parameter (left column). The reference to the corresponding figure is given in the middle column. CI, confidence interval; n, number of animals; n*, number of sections; n°, number of recordings.

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
