## [Decision Letter]

Thank you for submitting your article "Early tissue damage and microstructural reorganization predict disease severity in experimental epilepsy" for consideration by *eLife*. Your article has been reviewed by two peer reviewers, and the evaluation has been overseen by a Reviewing Editor and Sabine Kastner as the Senior Editor. The reviewers have opted to remain anonymous.

The reviewers have discussed the reviews with one another and the Reviewing Editor has drafted this decision to help you prepare a revised submission.

Summary:

This manuscript addresses the important issue of the relationship between early neuronal damage and subsequent development of temporal lobe epilepsy in a rodent animal model of unilateral focal hippocampal injury. The experiments are well designed and the results well illustrated and carefully discussed. Multimodal imaging methods are used to determine early changes associated with the epileptogenic phase, and correlate these with cellular/tissue level changes. Correspondence is also made with similar imaging approaches in human epileptic patients. Overall these results suggest that imaging biomarkers might be critically useful biomarkers for the epileptogenic state, fulfilling an important clinical need.

Essential revisions:

Epileptogenesis: The intrahippocampal KA model is technically challenging to master. In contrast to most TLE other models, there is electrographic activity in the hippocampus throughout the early epileptogenic period. Thus it can be difficult to clearly discriminate between the latent phase and the onset of the chronic phase, and this issue is not addressed. Were the electrographic characteristics of the latent and chronic phase distinct enough to discriminate between epileptogenesis and epilepsy? Please clarify these points in the Results and Discussion. Further, it would be important to reproduce these results in other models such as like the systemic KA or pilocarpine systemic model in which the onset of the chronic phase is clearer. At the very least this potential limitation needs to be adequately discussed.

Hippocampus target focus: Previous studies using MRI have shown that signal changes better predicting epilepsy are not located in hippocampus but in piriform cortex (pilocarpine) or amygdala (febrile seizures). The authors should justify why they concentrated only on hippocampus.

Biomarkers: The authors propose that their findings allowed the identification of a new biomarker for predicting epileptogenesis and possibly the severity of the disease and this approach could be translated to the human disease. How would they propose to do so and how would they select their population at risk, which criteria, TBI, SE, FS, etc…?

Metabolite measurement: Metabolite concentrations were determined using water as reference, yet significant edema was present in the tissue as indicated by T2 contrast. Please estimate how much that could explain the detected apparent decrease in metabolite concentrations.

DG dispersion: The authors describe mossy fiber sprouting and reorganization of granule cells as potential contributing factors to Diffusion changes in DG. However, a recent study shows that myelinated axons in outer molecular layer may also contribute (Laitinen et al. Neuroimage. 2010 Jun;51(2):521-30). This should be discussed.

Discussion of DWI: The authors state: "DWI successfully identifies changes of water diffusion in several rodent epilepsy modes. However, in these studies detailed histological correlation with DWI metrics is lacking". This statement is incorrect. There is a significant body of work combining high resolution diffusion MRI in hippocampus with histology in epilepsy models (see e.g. Kuo et al. Kuo Neuroimage. 2008 Jul 1;41(3):789-800; Laitinen et al. Neuroimage. 2010 Jun;51(2):521-30. Salo RA et al. Neuroimage. 2017 Mar 4;152:221-236, Sierra 2015b).

DTI methods: Voxel size in DTI is far from isotropic (spaghetti voxel). Also b-value of 1000 s/mm^2^ can be considered relatively small for approaches going beyond conventional DTI. Please discuss influence of these on DTI and especially fiber tracking approaches.

[Editors' note: further revisions were requested prior to acceptance, as described below.]

Thank you for resubmitting your work entitled "Early tissue damage and microstructural reorganization predict disease severity in experimental epilepsy" for further consideration at *eLife*. Your revised article has been favorably evaluated by Sabine Kastner (Senior Editor), and a Reviewing Editor.

The manuscript has been improved but there is one remaining issue that needs to be addressed before acceptance, as outlined below:

The one remaining issue is that the reporting on epileptogenesis (Figure 1—figure supplement 1) is anecdotal. No quantification is provided, nor any source data or statistical evidence to support the new findings described in the last paragraph of the subsection “Inter-individual variability of histological changes associated with mTLE”.

---

## [Author Response]

Essential revisions:

Epileptogenesis: The intrahippocampal KA model is technically challenging to master. In contrast to most TLE other models, there is electrographic activity in the hippocampus throughout the early epileptogenic period. Thus it can be difficult to clearly discriminate between the latent phase and the onset of the chronic phase, and this issue is not addressed. Were the electrographic characteristics of the latent and chronic phase distinct enough to discriminate between epileptogenesis and epilepsy? Please clarify these points in the Results and Discussion. Further, it would be important to reproduce these results in other models such as like the systemic KA or pilocarpine systemic model in which the onset of the chronic phase is clearer. At the very least this potential limitation needs to be adequately discussed.

We follow the reviewer's opinion, that the issue of discriminating between the latent phase and chronic epilepsy was not sufficiently addressed in the former manuscript. Given that chronically implanted electrodes would massively interfere with MRI measurements, we performed new experiments in another set of animals investigating the time-course of neuronal activity during the first two weeks following SE, the time span which is typically considered as the latent phase in the intrahippocampal KA mouse model. Our new data is consistent with previous observations made in the same model, showing a development from mainly isolated epileptic spikes and low-amplitude bursts to prolonged high-amplitude recurrent paroxysmal discharges over a period of approximately two weeks (Riban et al., Neuroscience, 2002, 112(1):101-11; Arabadzisz et al., Exp Neurol, 2005, Jul;194(1):76-90; Heinrich et al., Neurobiol Dis, 2011, Apr;42(1):35-47). Therefore, we consider the emergence of recurrent paroxysmal discharges at around two weeks after SE as the onset of the chronic phase. These results are presented (subsection “Inter-individual variability of histological changes associated with mTLE”, last paragraph; Figure 1—figure supplement 1) and discussed (Discussion, first, fourth paragraphs), and the Methods section was adapted (subsection “Animals”; subsection “Electrode implantation and local field potential recordings”). We also rephrased potentially misleading parts in the Introduction (last paragraph) and the Discussion (first paragraph). Moreover, in the Discussion we extended the comparison of our biomarkers with findings made in systemic epilepsy models to better delineate the reproducability of the identified biomarkers (second and fourth paragraphs).

Hippocampus target focus: Previous studies using MRI have shown that signal changes better predicting epilepsy are not located in hippocampus but in piriform cortex (pilocarpine) or amygdala (febrile seizures). The authors should justify why they concentrated only on hippocampus.

In the present study, we focused on the hippocampus since the sclerotic hippocampus represent the most common epileptogenic focus in drug-resistent mTLE (Malmgren and Thom, Epilepsia, 2012, Sep;53 Suppl 4:19-33; Cendes et al., Acta Neuropathol, 2014, Jul;128(1):21-37; Walker, Semin Neurol, 2015, Jun;35(3):193-200). A corresponding statement can be found in the revised Introduction (last paragraph). However, we appreciate the reviewer's impulse to include also other limbic brain regions (i.e. the piriform cortex and the amygdala) which are known to be vulnerable to epileptogenic insults. Therefore, we also analyzed T2 and DWI changes in the piriform cortex and the amygdala. Our findings are presented (subsection “Early neuronal cell death determines disease severity in chronic epilepsy”, second paragraph;subsection “Microstructural alterations during intermediate epileptogenesis correspond to disease progression"; Figure 2—figure supplement 1; Figure 5—figure supplement 2) and discussed accordingly (Discussion, fourth paragraph).

Biomarkers: The authors propose that their findings allowed the identification of a new biomarker for predicting epileptogenesis and possibly the severity of the disease and this approach could be translated to the human disease. How would they propose to do so and how would they select their population at risk, which criteria, TBI, SE, FS, etc…?

We agree with the reviewer's suggestion and now propose an experimental design for a translational approach in the revised Discussion (last paragraph). To illustrate this approach we added a schematic to the supplementary figures (Figure 9—figure supplement 1).

Metabolite measurement: Metabolite concentrations were determined using water as reference, yet significant edema was present in the tissue as indicated by T2 contrast. Please estimate how much that could explain the detected apparent decrease in metabolite concentrations.

A common practice is to use total creatine as reference, but we refrained from this as there is no guarantee that the creatine concentration remains constant during the course of epileptogenesis. On the contrary, massive cell loss after SE inevitably leads to reduced concentrations of many metabolites. Therefore it was decided to use water for absolute quantification. The reviewer is right that several processes in the septal HC – cell death, edema and gliosis – gave rise to increased signal intensities in the T2-weighted images. Certainly these effects might also influence the water reference for the metabolite measurements. As a quantitative assessment of this is hardly possible, we can only qualitatively consider how much that influenced the metabolite quantification. The signal intensity in the images went back to baseline between the time-points day one and four after SE. As all metabolites were scaled by the same reference, this trend – increase at day 1 and decrease at day 4 – would have also been visible in all metabolites if it was purely due to changes in the water reference. Looking at several spectra we can reject this hypothesis. For instance, glutamine and alanine revealed no significant changes, creatine decreased but remained decreased and phosphocreatine decreased but returned to baseline not until day 8 (see Figure 10). Therefore we can conclude that, although changes of the water concentration might have a small influence on the metabolite quantification, the significant portion of the concentration changes reflect the described metabolic processes. A strategy for a better quantification in the future would be to take the contralateral water concentration as reference. But this approach requires the acquisition using a volume coil, as a surface coil (like the used two-element cryo-probe) leads to different intensities depending on the loading of the respective coil-element. We additionally show (Figure 10) some examples of longitudinal metabolite concentrations relative to creatine instead of water, demonstrating that the observed trends were also present independently of water. We have now summarized the above points in the Methods (subsection **“**Methodological considerations of MR acquisitions”, first two paragraphs).

Author response image 1.**DOI:**
http://dx.doi.org/10.7554/eLife.25742.025

DG dispersion: The authors describe mossy fiber sprouting and reorganization of granule cells as potential contributing factors to Diffusion changes in DG. However, a recent study shows that myelinated axons in outer molecular layer may also contribute (Laitinen et al. Neuroimage. 2010 Jun;51(2):521-30). This should be discussed.

Although we have not addressed the reorganization of myelinated fibers in the present study, we agree that this could also contribute to diffusion changes in the DG. We therefore added this point to the Discussion (fifth paragraph).

Discussion of DWI: The authors state: "DWI successfully identifies changes of water diffusion in several rodent epilepsy modes. However, in these studies detailed histological correlation with DWI metrics is lacking". This statement is incorrect. There is a significant body of work combining high resolution diffusion MRI in hippocampus with histology in epilepsy models (see e.g. Kuo et al. Kuo Neuroimage. 2008 Jul 1;41(3):789-800; Laitinen et al. Neuroimage. 2010 Jun;51(2):521-30. Salo RA et al. Neuroimage. 2017 Mar 4;152:221-236, Sierra 2015b).

We understand that our statement is framed too drastically and does not necessarily give justice particularly to the studies the reviewer quoted. We rephrased our statement accordingly (Discussion, fourth paragraph).

DTI methods: Voxel size in DTI is far from isotropic (spaghetti voxel). Also b-value of 1000 s/mm^2^ can be considered relatively small for approaches going beyond conventional DTI. Please discuss influence of these on DTI and especially fiber tracking approaches.

In diffusion measurements there is always a tradeoff between desired features – high resolution, strong diffusion weighting (b-value) and many diffusion directions – and the resulting acquisition time and signal-to-noise ratio (SNR). In our ex-vivo measurements of human hippocampal tissue, acquisition time was less critical permitting a 3D scan with 200µm^3^ isotropic resolution, b-values up to 5000s/mm^2^ and 50 diffusion directions. For in-vivo measurements time is critical and we had to compromise. This study focused on alterations within the hippocampus. In the transverse axis different layers within the hippocampus reveal much finer structures as compared to the longitudinal axis. Consequently, we also utilized an anisotropic voxel geometry. Choosing an isotropic voxel geometry with the same in-plane resolution would drastically reduce the SNR while adding comparatively little information. It has even been shown that in DTI an anisotropic analysis, averaging noise along a rather homogeneous direction and reducing partial volume effects, could lead to increased sensitivity of detecting a pathology (Van Hecke et al., Hum Brain Map*p*, 2009, Jan;31(1):98-114). For in-vivo DTI a b-value of 1000s/mm^2^ and 30 diffusion directions is in the optimal range (Derek Jones; Diffusion MRI: Theory, Methods, and Applications; Oxford University Press, 2012). Although fiber tracking approaches generally benefit from higher b-values und more diffusion directions, higher b-values would have led to low SNR and several studies have validated the results of the applied tractography method using similar acquisition schemes [Harsan et al., PNAS, 2009, May 7;110(19):E1797-806; Anastasopoulos et al., AJNR Am J Neuroradiol, 2014, Feb;35(2):291-6; Horn et al., Neuroimage, 2014, Nov 15;102 Pt 1:142-51]. We have added a section in the Methods considering these issues (subsection “Methodological considerations of MR acquisitions”, last paragraph).

[Editors' note: further revisions were requested prior to acceptance, as described below.]

The manuscript has been improved but there is one remaining issue that needs to be addressed before acceptance, as outlined below:

The one remaining issue is that the reporting on epileptogenesis (Figure 1—figure supplement 1) is anecdotal. No quantification is provided, nor any source data or statistical evidence to support the new findings described in the last paragraph of the subsection “Inter-individual variability of histological changes associated with mTLE”.

We followed your request and performed a quantitative analysis of our longitudinal EEG data. We added these results to the revised manuscript (subsection “Inter-individual variability of histological changes associated with mTLE”, last paragraph; Figure 1—figure supplement 1) and described the analysis in the Methods section (subsection “Electrode implantation and local field potential recordings”, first paragraph). The detailed results of the statistical testing have also been added to [Supplementary-material SD5-data].